# Phase separation of PGL-3 driven by structured domains that oligomerize and interact with RGG motifs

Rimpei Kuroiwa [1], Piyoosh Sharma [2,4], Andrea A Putnam [1,5], Stephen D Fried [2,3] & Geraldine Seydoux [1✉]

## Abstract

Phase separation (PS) of biomolecular condensates is often assumed to be driven by interactions involving nucleic acids and intrinsically disordered regions (IDRs) of proteins. PGL-3 is a component of P granules, biomolecular condensates in *C. elegans*, that contains two structured domains (D1-D2), an internal IDR, and a C-terminal IDR rich in RGG motifs. Theoretical and in vitro studies implicated the internal IDR and RGG motifs in driving PGL-3 PS via self-interactions and binding to RNA. Studies in cells, however, implicated the D1 and D2 domains. Here, we investigate the molecular basis of PGL-3 PS in vitro using microscopy, crosslinking mass spectrometry, and biophysical measurements. We find that D1-D2 forms oligomers and is necessary and sufficient for PS. The terminal RGG region interacts with D1-D2 in a manner that enhances PS even in the absence of RNA. In contrast, the internal IDR is neither necessary nor sufficient for PS. These findings support an alternative model for PGL-3 PS that does not require RNA and is driven by oligomerization of structured domains that interact with RGG repeats.

Keywords Phase separation; oligomerization; RGG motifs; bimolecular condensates; intrinsically disordered region
Subject Category RNA Biology

## Introduction

Biomolecular condensates are proposed to assemble by liquid-liquid phase separation (LLPS), a thermodynamically driven de-mixing phenomenon (Boeynaems et al, 2018; Banani et al, 2017; Hyman et al, 2014). Several studies have highlighted the role of nucleic acids and intrinsically disordered regions (IDRs) of proteins in driving phase separation. Studies focused on FUS and TDP-43 revealed that low-complexity regions (LCRs), a subclass of IDRs,

provide weak, multivalent interactions, including cation–π interactions and π–π stacking, that drive LLPS (Portz et al, 2021). These studies have motivated the development of predictors that estimate LLPS propensity based on primary sequence with no or limited three-dimensional structural information (Hardenberg et al, 2020; Saar et al, 2021; Sun et al, 2024; Liang et al, 2024; Hou et al, 2024; Vendruscolo and Fuxreiter, 2022; Ibrahim et al, 2023).

Phase separation of IDR-containing proteins can also involve folded domains (Uversky, 2022). For example, TDP-43 condensation is enhanced by the head-to-tail oligomerization of its N-terminal domain (Wang et al, 2018; Carter et al, 2021). Folded domains can also be binding partners for short linear motifs (SLiMs) in IDRs (Hess and Joseph, 2025), as in the case of Rubisco, which phase separates with the disordered EPYC1 protein in pyrenoids (Meyer et al, 2020; He et al, 2020; Atkinson et al, 2019), and NCK, whose SH2 and SH3 domains interact with proline-rich motifs (PRMs) in N-WASP and pTyr residues in Nephrin, respectively (Li et al, 2012; Banjade and Rosen, 2014). Synthetic reconstitutions have also shown that dimerization and oligomerization induced by folded domains can drive LLPS by bringing together self-interacting multivalent IDRs (Garabedian et al, 2022; Shin et al, 2017; Bracha et al, 2018). In the case of G3BP, an N-terminal dimerization domain and a C-terminal RNA-binding domain collaborate to drive phase separation in the presence of RNA (Yang et al, 2020).

Whether structured domains can drive phase separation in the absence of IDRs or nucleic acids is less clear. A molecular dynamics simulation study reported that two multimeric coiled-coil (CC) domains connected by a linker segment would be sufficient for phase separation (Ramirez et al, 2024). SPD-5, a *C. elegans* pericentriolar material (PCM) scaffold protein that contains nine CC domains separated by mostly disordered linkers, can form glass-like or gel-like condensates in the presence of a crowding agent in vitro (Woodruff et al, 2017). Two coiled-coil (CC) domains in the *Drosophila* homolog of SPD-5 (centrosomin) were reported to form micron-scale assemblies when mixed in vitro (Feng et al, 2017). Similarly, a synthetic study showed that two CC domains fused to GFP by flexible linkers can form liquid-like condensates in cells and in the presence of a molecular crowder in vitro (Hilditch et al, 2024). These examples suggest that certain folded domains can drive condensation on their own.

[1]Department of Molecular Biology and Genetics, Johns Hopkins University, Howard Hughes Medical Institute, Baltimore, MD 21205, USA. [2]Department of Chemistry, Johns Hopkins University, Baltimore, MD 21218, USA. [3]Thomas C. Jenkins Department of Biophysics, Johns Hopkins University, Baltimore, MD 21218, USA. [4]Present address: Department of Biomedical Engineering, Washington University in Saint Louis, Saint Louis, MO 63130, USA. [5]Present address: Department of Biomolecular Chemistry, University of Wisconsin-Madison, Madison, WI 53706, USA. ✉E-mail: gseydoux@jhmi.edu

PGL proteins were the first proteins identified as constitutive components of P granules, biomolecular condensates in the *C. elegans* germline (Kawasaki et al, 1998; Brangwynne et al, 2009). Early studies reported that PGL-1 and its paralog PGL-3 are required for P granule assembly in embryos and form cytoplasmic granules when expressed individually or together in CHO cells (Hanazawa et al, 2011; Aoki et al, 2021). Crystallography and size exclusion chromatography of residues 1–212 and residues 205–447 in PGL-1 revealed that each form well-folded homodimerization domains (D1 and D2), separated by a likely disordered six amino-acid linker (Aoki et al, 2016, 2021). PGL-1 and PGL-3 are homologous with 60% identity overall and contain, in addition to the N-terminal D1 and D2 domains, an internal IDR (223 residues in PGL-1, and 175 residues in PGL-3) followed by a C-terminal, low sequence-complexity domain rich in arginine (R) and glycines (G), including 10 RGG and 1 RG motifs in PGL-1 and 6 RGG and 6 RG motifs in PGL-3 (Fig. 1A; Appendix Fig. S1A,B). We refer to this C-terminal domain as the "RGG region".

Different studies have pointed to different PGL domains driving condensation. A deletion spanning D1 and D2 prevented PGL-3 condensation in CHO cells and *C. elegans* embryos (Hanazawa et al, 2011), and point mutations in the D1 dimerization interface prevented PGL-1 condensation in CHO cells and in the *C. elegans* germline (Aoki et al, 2021). In addition, a preprint reported that the first 452 amino acids of PGL-3 (spanning D1 and D2) were sufficient for phase separation in vitro and that a C-terminal fragment (residues 370–693) failed to form condensates (preprint: Jelenic et al, 2024). A theoretical study, however, found that a region spanning a portion of the internal IDR and the RGG region (residues 515–693) was sufficient to reproduce the experimental phase diagram of full-length PGL-3 across varying temperatures and PGL-3 and salt concentrations (Meca et al, 2023). Finally, a study using recombinant PGL-3::GFP showed that mRNA enhances PGL-3 condensation in a manner dependent on the RGG region, which could bind RNA in vitro (Saha et al, 2016). Deletion of the RGG region, however, did not prevent PGL-3 condensation in transfected CHO cells and *C. elegans* embryos (Hanazawa et al, 2011). In other systems, disordered RGG motif-containing regions have been reported to mediate binding to nucleic acids (Kiledjian and Dreyfuss, 1992; Hanakahi et al, 1999; Takahama et al, 2011) and proteins, using homotypic RGG:RGG interactions (Shaw et al, 2010; Poornima et al, 2019) and heterotypic interactions involving other disordered domains (Yang et al, 2020; Murthy et al, 2021) or folded domains (Zhang and Cheng, 2003; Bourgeois et al, 2020; Kusakawa et al, 2007).

To complement these studies, we have undertaken a systematic analysis of PGL-3 phase separation in vitro to determine how each domain contributes to the phase separation properties of PGL-3 in the absence of RNA. We find that PGL-3 phase separation is driven primarily by oligomerization of D1 and D2 domains and tuned by interactions between the RGG region and D1-D2.

# Results

## Computational models predict high LLPS propensity for the IDR-RGG region of PGL-3

To investigate which regions in PGL-3 promote phase separation, we first analysed full-length PGL-3 and its sub-domains, using five

computational models trained to estimate LLPS propensity based on sequence (FuzDrop (Hardenberg et al, 2020), DeePhase (Saar et al, 2021), PSPHunter (Sun et al, 2024), MolPhase (Liang et al, 2024), PSPire (Hou et al, 2024)). All five models use features derived from the primary sequence for prediction. PSPire, in addition, considers surface-exposed residues in structured domains. The five models gave full-length PGL-3 a high propensity score for phase separation (0.7 or higher, where 0.5 is the threshold for binary prediction) (Fig. 1B). Comparing sub-regions of PGL-3, the models scored the IDR-RGG region highest and the D1-D2 domain lowest, consistent with the models relying primarily on protein disorder to predict phase separation.

## The D1-D2 domain is sufficient for phase separation

To experimentally determine the regions in PGL-3 responsible for phase separation, we turned to an in vitro reconstituted system. We purified from *E. coli* full-length, untagged PGL-3 and several derivatives and obtained a synthetic peptide spanning the RGG region (Methods; Figs. 1C and EV1A,B). Each protein preparation was trace-labeled (<1%) with Alexa Fluor 647 to facilitate condensate imaging by fluorescence microscopy. Condensation was examined in solutions containing 20 μM protein, 67.5 mM NaCl, 25 mM HEPES, pH 7.5, in the absence of RNA and crowding agents (Methods). Only full-length PGL-3 and derivatives that contained both the D1 and D2 domains yielded condensates (Fig.1C,D). Derivatives that contained only D1 or D2 failed to form condensates. Derivatives that contained the IDR and/or the RGG but lacked D1-D2 also failed to condense, even when tested at higher protein concentration (Fig. EV1C).

RNA enhances phase separation of full-length PGL-3 (Saha et al, 2016). As expected, when we supplemented the condensation reactions with RNA, we observed that RNA is enriched in condensates formed by full-length PGL-3 (Fig. 1E). RNA, however, did not enrich in D1-D2 condensates. The IDR-RGG region did not form condensates in the presence of RNA (Fig. 1E), even when tested at 100 μM (Fig. EV1D).

Aoki and co-workers (Aoki et al, 2021) reported that mutations (R123E, K126E, and K129E) that disrupt D1 dimerization block PGL-1 phase separation in tissue culture cells and in *C. elegans* germlines. We found that the same mutations introduced in full-length PGL-3 were sufficient to abrogate condensation at 10 and 30 μM protein and yielded only rare condensates when tested at 50 μM (Fig. EV1E). We conclude that the D1-D2 region of PGL-3 is necessary and sufficient for phase separation.

## D1-D2 exhibits a higher $c_{sat}$ and lower condensate viscosity compared to full-length PGL-3

We compared the phase diagram of full-length PGL-3 and D1-D2 across protein and salt concentrations using solution turbidity (Fig. 2A; Appendix Fig. S2A) and direct observation of condensates by microscopy (Fig. 2B; Appendix Fig. S2B). Both assays revealed that, at a given salt concentration, condensation of D1-D2 requires higher protein concentrations than full-length PGL-3. In a phase separation regime, the protein concentration in the dilute phase ($c_{dil}$) equals the saturation concentration ($c_{sat}$) and is independent of input concentration. This expectation is realized for both full-length PGL-3 and D1-D2, with D1-D2 exhibiting a higher $c_{dil}$

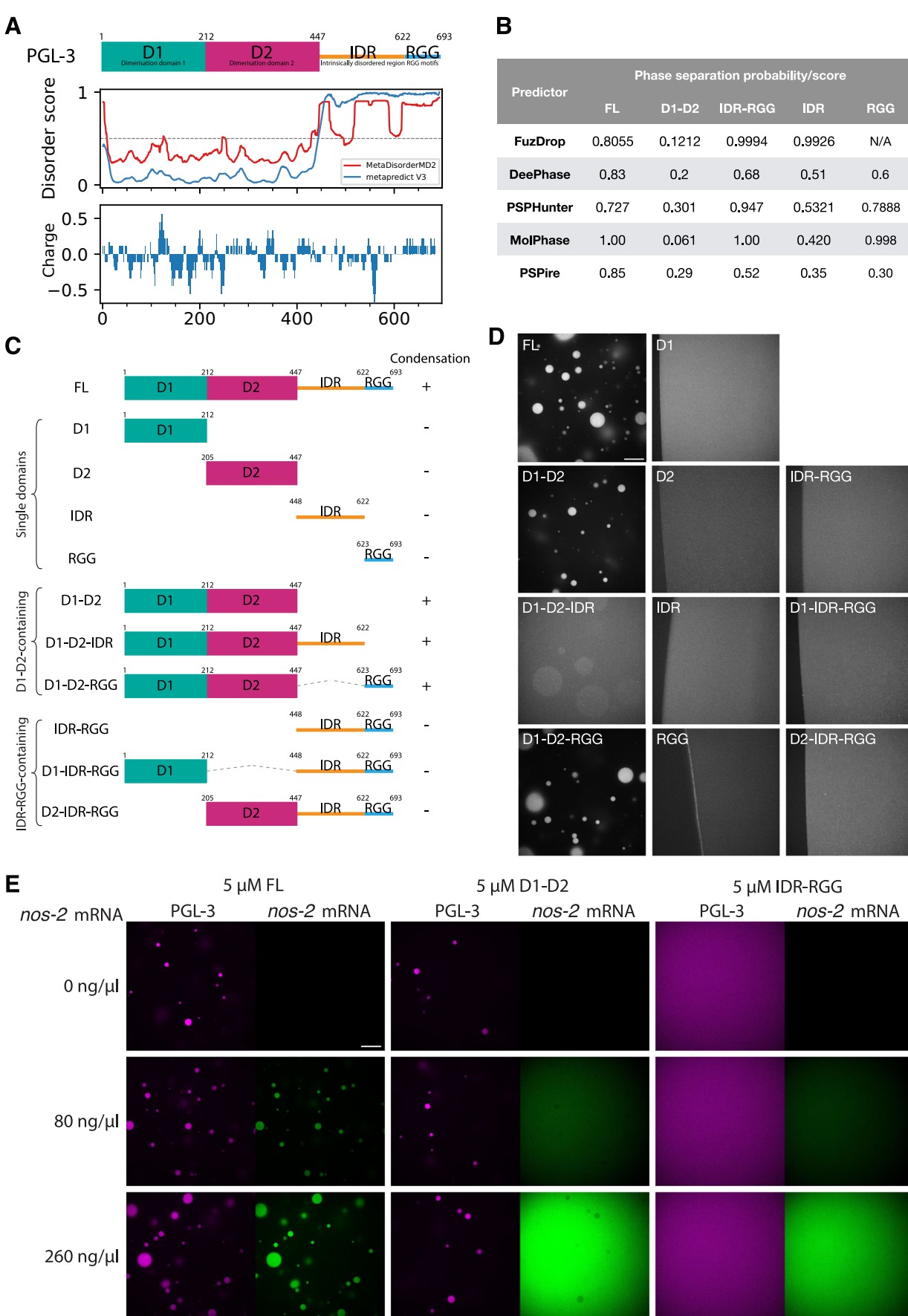

**B**

| Predictor | Phase separation probability/score | | | | |
|---|---|---|---|---|---|
| | FL | D1-D2 | IDR-RGG | IDR | RGG |
| FuzDrop | 0.8055 | 0.1212 | 0.9994 | 0.9926 | N/A |
| DeePhase | 0.83 | 0.2 | 0.68 | 0.51 | 0.6 |
| PSPHunter | 0.727 | 0.301 | 0.947 | 0.5321 | 0.7888 |
| MolPhase | 1.00 | 0.061 | 1.00 | 0.420 | 0.998 |
| PSPire | 0.85 | 0.29 | 0.52 | 0.35 | 0.30 |

◀ **Figure 1.** **The N-terminal D1-D2 domain is necessary and sufficient for phase separation.**

(A) Schematic showing the domain architecture of PGL-3 aligned to disorder scores predicted from MetaDisorderMD2 (Kozlowski and Bujnicki, 2012) and metapredict V3 (preprint: Lotthammer et al, 2024), and a charge plot by EMBOSS-charge (Madeira et al, 2024). (B) Table showing the phase separation probabilities predicted for full-length, the ordered region of PGL-3 (D1-D2) and the disordered region of PGL-3 (IDR-RGG), as well as IDR and RGG separately, using FuzDrop (Hardenberg et al, 2020), DeePhase (Saar et al, 2021), PSPHunter (Sun et al, 2024), MolPhase (Liang et al, 2024) and PSPire (Hou et al, 2024). The FuzDrop score for the RGG region shows N/A as FuzDrop requires a longer input sequence. (C) Schematics showing PGL-3 fragments tested for condensation in vitro as shown in (D). (D) Fluorescence micrographs of condensation reaction mixtures. The indicated PGL-3 domains (20 µM, 1% trace-labeled with Alexa Fluor 647) were mixed in 67.5 mM NaCl, 25 mM HEPES, pH 7.5 on a glass bottom dish at 19 °C for 10 min. In micrographs with no condensates, the air–water interface is shown on the left to highlight the homogeneous distribution of fluorescence in solution. Scale bar = 20 µm. (E) Fluorescence micrographs of 5 µM PGL-3 and derivatives (<1% trace-labeled with Alexa Fluor 647) with varying concentrations of *nos-2* mRNA (trace-labeled with DyLight 488) in 125 mM NaCl. To highlight the distribution of labeled components, fluorescence intensities were adjusted for each protein and are comparable only across each column (micrographs containing the same protein with varying amounts of RNA). Scale bar = 20 µm. Source data are available online for this figure.

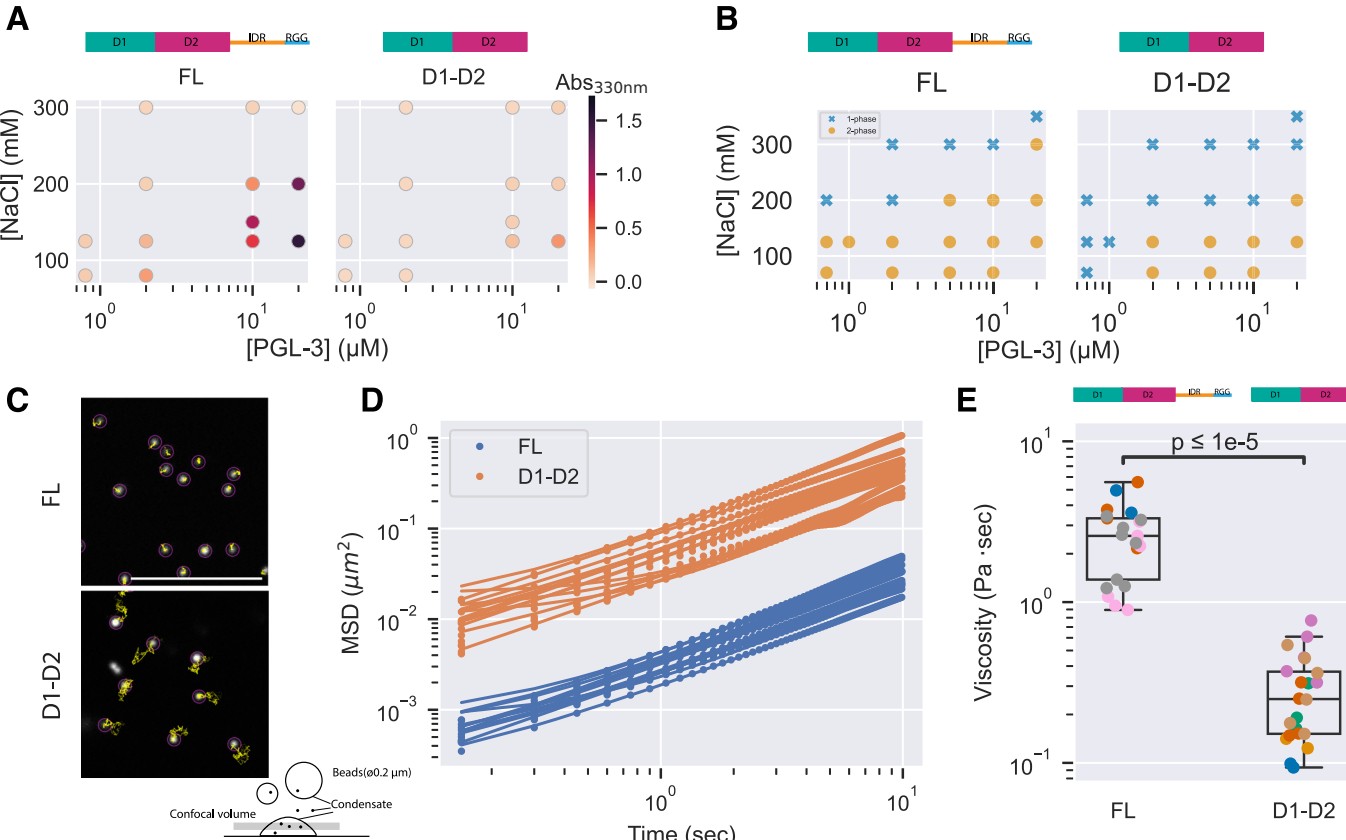

**Figure 2.** **Properties of condensates assembled with full-length PGL-3 or D1-D2.**

(A) Plots showing the turbidity (absorbance at 330 nm) of solutions containing full-length PGL-3 or D1-D2 at the indicated protein and salt concentrations in 25 mM HEPES, pH 7.5, <2.7% glycerol at 19 °C. Values shown are the average of three technical replicates. (B) Plots showing the presence (2-phase, dots) or absence (1-phase, crosses) of condensates visualized by fluorescent microscopy in solutions containing full-length PGL-3 or D1-D2 trace-labeled with Alexa Fluor 647 in 25 mM HEPES, pH 7.5, <2.7% glycerol at 19 °C. (C) Micrographs showing the trajectories over 15 s of polystyrene beads (0.2 µm) embedded inside a full-length PGL-3 or D1-D2 condensate. Each image shows a close-up of beads inside one condensate. Magenta circles and yellow lines indicate bead positions and trajectories, respectively. Scale bar is 5 µm. (D) Plot showing the mean-squared displacement (MSD) of beads diffusing inside full-length PGL-3 or D1-D2 condensates. Each line corresponds to the mean of MSDs from one condensate. Technical replicates. $N > 20$ condensates. (E) Plot showing the estimated viscosity based on measurements shown in (C, D). Diffusion coefficients calculated from MSD were applied to the Stokes–Einstein–Sutherland equation to estimate the viscosity. Each dot represents a single condensate; colors designate condensates examined on the same condensation reaction. In the overlaying box plots, lines inside boxes show median (Q2), box bounds are quartiles (Q1 and Q3), and whiskers show the range of data excluding outliers identified by the Turkey method. *P* value ($2.139 \times 10^{-8}$) calculated by the Wilcoxon rank-sum test. Technical replicates. $N > 20$ condensates. Source data are available online for this figure.

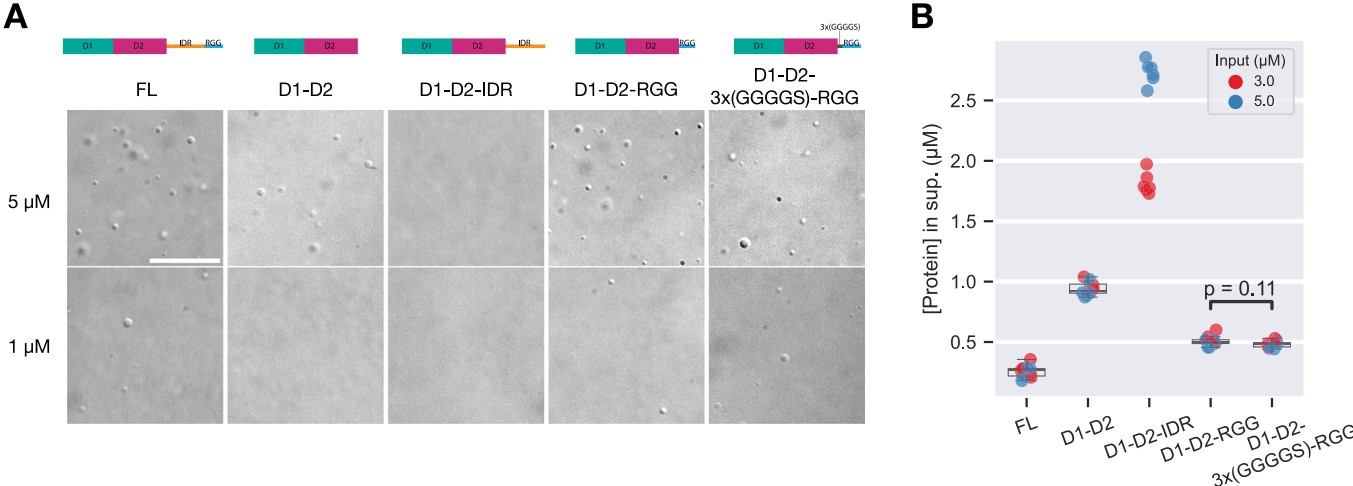

**Figure 3. The RGG region enhances D1-D2 condensation.**

(A) DIC micrographs of solutions containing FL and D1-D2-containing derivatives of PGL-3 at 1 or 5 μM, 125 mM NaCl. Scale bar = 20 μm. (B) Plot showing the measured protein concentrations of the dilute phase with 3 μM (red) and 5 μM (blue) input PGL-3 concentrations in 125 mM NaCl. PGL-3 condensation reactions were incubated and centrifuged to pellet the dense phase. The supernatant was used to estimate the $c_{sat}$ by BCA assay. In the overlaying box plots, lines inside boxes show median (Q2), box bounds are quartiles (Q1 and Q3), and whiskers show the range of data excluding outliers identified by the Turkey method. *P* values calculated by Wilcoxon rank-sum test with Benjamini–Hochberg adjustment for multiple comparisons. Technical replicates. *N* = 6. *P* value = 0.112 (D1-D2 vs. D1-D2-(3xGGGGS)-RRG), $3.602 \times 10^{-5}$ (D1-D2 vs. D1-D2-RGG), $3.602 \times 10^{-5}$ (FL vs. D1-D2-RGG), $3.630 \times 10^{-5}$ (D1-D2 vs. D1-D2-(3xGGGGS)-RGG) and $3.630 \times 10^{-5}$ (FL vs. D1-D2-(3xGGGGS)-RGG). Source data are available online for this figure.

compared to full-length PGL-3 at the three salt concentrations tested (Fig. EV2A). At 125 mM NaCl, we observed $c_{dil}$ of ~0.25 and ~1 μM for full-length PGL-3 and D1-D2, respectively. Consistent with the higher $c_{sat}$, D1-D2 condensates were moderately smaller and less abundant than full-length PGL-3 condensates (Figs. EV2B,C; Appendix Fig. S2C).

To estimate viscosity, we embedded fluorescent beads in the condensates and measured their diffusion by single particle tracking (Methods). Beads in D1-D2 condensates exhibited larger mean square displacements (MSD) compared to beads in full-length PGL-3 condensates (Fig. 2C,D; Movie EV1). Using the Stokes–Einstein–Sutherland equation ($D = k_B T / 6\pi\eta a$), we estimated the viscosity of full-length PGL-3 condensates to be $\eta = 3.30 \pm 0.57$ Pa·s, consistent with values previously reported in the literature (Brangwynne et al, 2009; Jawerth et al, 2020; Folkmann et al, 2021), compared to $0.25 \pm 0.09$ Pa·s for D1-D2 condensates (Fig. 2E). We confirmed this observation by measuring fluorescence recovery after photobleaching (FRAP) of full-length and D1-D2 condensates. Consistent with the microrheology data, full-length condensates recovered more slowly than D1-D2 condensates (Fig. EV2D,E). We conclude that D1-D2 condensates are one order of magnitude less viscous than full-length PGL-3 condensates, suggesting that the IDR and/or RGG region modulates the properties of PGL-3 condensates.

## The RGG region enhances PGL-3 condensation

To examine possible contributions of the IDR and RGG region to phase separation directly, we compared $c_{sat}$ estimates for derivatives that contain D1-D2 with the IDR or RGG region (Methods). At 125 mM NaCl and 5 μM PGL-3, D1-D2 formed droplets, but D1-D2-IDR did not (Fig. 3A). $C_{dil}$ of D1-D2-IDR varied

proportionally to input protein concentrations, confirming lack of phase separation (Fig. 3B; the apparently lower $c_{dil}$ values are likely due to adsorption to tube surfaces). We conclude that the internal IDR reduces the phase separation propensity of D1-D2 when present in the absence of the RGG region.

In contrast, we found that the RGG region enhanced phase separation even in the absence of the internal IDR. D1-D2-RGG formed condensates at 1 and 5 μM protein in 125 mM NaCl (Fig. 3A). The $c_{dil}$ of D1-D2-RGG was $0.51 \pm 0.02$ μM, lower than D1-D2 ($0.94 \pm 0.03$ μM) (Fig. 3B; Appendix Fig. S3) but higher than full-length PGL-3's $c_{dil}$ ($0.26 \pm 0.03$ μM). Inserting a 3x(GGGGS) flexible linker between D1-D2 and the RGG region did not significantly change $c_{dil}$ (Fig. 3A,B), possibly due to its shorter length (15 residues) compared to the internal IDR (175 residues). We conclude that the RGG region enhances condensation of D1-D2.

## Crosslinking mass spectrometry reveals multiple interactions between the RGG region and D1 and D2

To directly assess molecular interactions that accompany PGL-3 condensation, we performed crosslinking mass spectrometry (XL-MS) under salt regimes permissive and non-permissive for condensation (125 and 500 mM NaCl, respectively). We classified crosslinks within D1 and D2 into five categories based on their compatibility with SWISS-MODEL (Waterhouse et al, 2018) models of the PGL-3 D1 and D2 domains templated on the published dimer structures of isolated PGL-1 D1 and isolated PGL-1 D2 (Fig. EV3A) (Aoki et al, 2016, 2021), as well as the monomer structures extracted from these. Crosslinks were considered "compatible" with monomer or dimer structures (intra-chain or inter-chain, respectively) if alpha carbons of the corresponding residues were less than 30 Å apart in those models (Methods).

We detected several homodimer crosslinks (Fig. 4A, downward-pointing loops), which we define as crosslinks that occurred between the same residue on peptide-pairs that overlap in sequence. Because each tryptic PGL-3 fragment is unique, homodimer crosslinks must derive from two interacting PGL-3 molecules. D1:D1 homodimer crosslinks at K134 and D2:D2 homodimer crosslinks at K352 were compatible with the published D1 and D2 dimer structures (Aoki et al, 2016, 2021) (Fig. 4A, yellow downwards loop crosslinks). We found other D1:D1 and D2:D2 homodimer crosslinks that were incompatible with published dimer structures: K5 and K277 (observed in both conditions), S272 (observed only under the condensing condition), and S434 (observed only under the non-condensing condition) (Figs. 4A and EV3B, magenta crosslinks). These crosslinks suggest that the D1 and D2 domains contain at least one additional homotypic binding interface.

We also detected many crosslinks between peptides in different domains of PGL-3 ("inter-domain" crosslinks) (Fig. 4A). Unlike homodimer crosslinks, it is not possible to distinguish whether inter-domain crosslinks occurred intra- or inter-molecularly. We detected 18 spectra corresponding to crosslinks that connect residues within the IDR-RGG region and 58 spectra corresponding to crosslinks that connect residues in the D1-D2 region to residues in the IDR-RGG region. Among the latter, 55 involved the RGG region and only three involved the IDR (Fig. 4B). Remarkably, all the crosslinks between D1 and the RGG region appeared only under condensing conditions (Fig. 4A). We systematically compared the frequency of each crosslink class under the two salt concentrations (Fig. 4C). This analysis revealed that D1:RGG and D1:D1 crosslinks are favored under condensing conditions. In contrast, D2:D2, D2:RGG, IDR:RGG and RGG:RGG crosslinks are favored under non-condensing conditions (Fig. 4C). Crosslinks involving the IDR are the rarest crosslink type.

To explore whether the homodimer crosslinks incompatible with published dimer structures can be explained by higher oligomers, we generated D1-D2 oligomer models using Alpha-Fold3. A tetramer model for D1-D2 showed two K5 residues within 30 Å, but S272 and K277 remained further apart, inconsistent with the crosslink data (Appendix Fig. S4A,B). The D1 configuration in the tetramer model scored medium to high paired alignment errors (PAE) (Appendix Fig. S4C, magenta squares), indicating that the model is speculative. Higher-order oligomer models, including some compatible with S272 and K277 dimer crosslinks, also yielded moderate to high pair alignment errors for the novel interfaces.

The XL-MS findings suggest that the D1 and D2 domains are multivalent, possibly forming oligomers, and interact with the RGG region. The internal IDR, in contrast, rarely participates in intermolecular interactions either with itself or with other PGL-3 domains.

## D1 and D2 form oligomers and interact with the RGG region

To directly test whether D1-D2 forms oligomers, we examined dilute solutions (50 to 100 nM) of D1-D2 by mass photometry, an interferometric scattering based technique that allows estimation of the mass of molecular complexes in solution (Fig. EV4A). We readily detected particles whose masses match the theoretical sizes of D1-D2 monomers, dimers, tetramers and hexamers, even at tens of nano-molar concentration range, well below the physiological concentration of PGL-3 (0.68 μM (Saha et al, 2016)). As expected, the frequency of larger oligomers increased with concentration. These observations confirm that D1-D2 can form oligomers, consistent with work by Aoki and co-workers (Aoki et al, 2016), who observed D1-D2 trimers and/or tetramers using chemical crosslinking.

Individual D1 or D2 domains are too small to resolve reliably by mass photometry. Therefore, we used blue-native PAGE to attempt to resolve native oligomers (Fig. EV4B). We found that D1 migrates mostly as monomers but shows extensive smearing, indicative of transient complex formation. The D1 smear extends beyond the expected dimer size (~46 kDa). D2 migrates mostly as monomers and dimers and also shows faint higher molecular weight bands and smears, again suggesting unstable complexes. These data suggest that D1 and D2 form transient oligomers, which would explain our inability to detect high molecular weight oligomers by mass photometry.

To stabilize oligomers and better examine their oligomerization potential, we turned to chemical crosslinking using 1% formaldehyde (FA) and trace-labeled D1 and D2 (DL488). The crosslinked species were resolved by SDS-PAGE and examined by in-gel fluorescent imaging (Methods). We found that both D1 and D2 formed dimers and higher oligomeric complexes, confirming that these domains can form oligomers when present as single domains in solution (Fig. 5A).

To determine whether D1 or D2 also interact with the IDR or RGG region, we trace-labeled the IDR and RGG region with a different fluorophore [Alexa Fluor 674 (AF647)] to detect possible hetero-oligomers. We also tested a KGG peptide where all the Arg residues in the RGG region were replaced with Lys (Fig. EV4C). Mixing the internal IDR with D1 or D2 did not affect the migration patterns of either species, confirming minimal interactions between these domains (Fig. 5A). In contrast, mixing D1 with RGG led to a dramatic shift in the migration patterns of both D1 (labeled with DL488) and RGG (labeled with AF647). The novel bands were positive for both fluorophores, consistent with D1:RGG crosslinked species (Figs. 5A Right and EV4D). The apparent sizes of the crosslinked D1:RGG species were consistent with 1:1, 1:2, 1:3, and 2:1 stoichiometries, as well as higher-order oligomers of unknown stoichiometries. Similarly, we also detected D2:RGG crosslinked species. Most of the heterotypic species were absent in 500 mM NaCl (Fig. 5A), suggestive of electrostatic interactions. We also tested a scrambled RGG peptide in which all six RGG triplets were disrupted (Fig. EV4C; Methods). The scrambled RGG peptide crosslinked to D1 and D2 as efficiently as the native RGG peptide (Appendix Fig. S5A), supporting the theory that the interaction is largely electrostatic. We conclude that that the RGG region interacts with both the D1 and D2 domains, as also observed in the XL-MS data.

## The RGG region enriches in D1-D2 condensates when provided in *trans*

To examine whether D1-D2 condensates can recruit the RGG peptide in *trans*, we induced D1-D2 condensation in the presence of equimolar amounts of IDR, RGG and KGG peptides and determined their partition coefficients in D1-D2 condensates

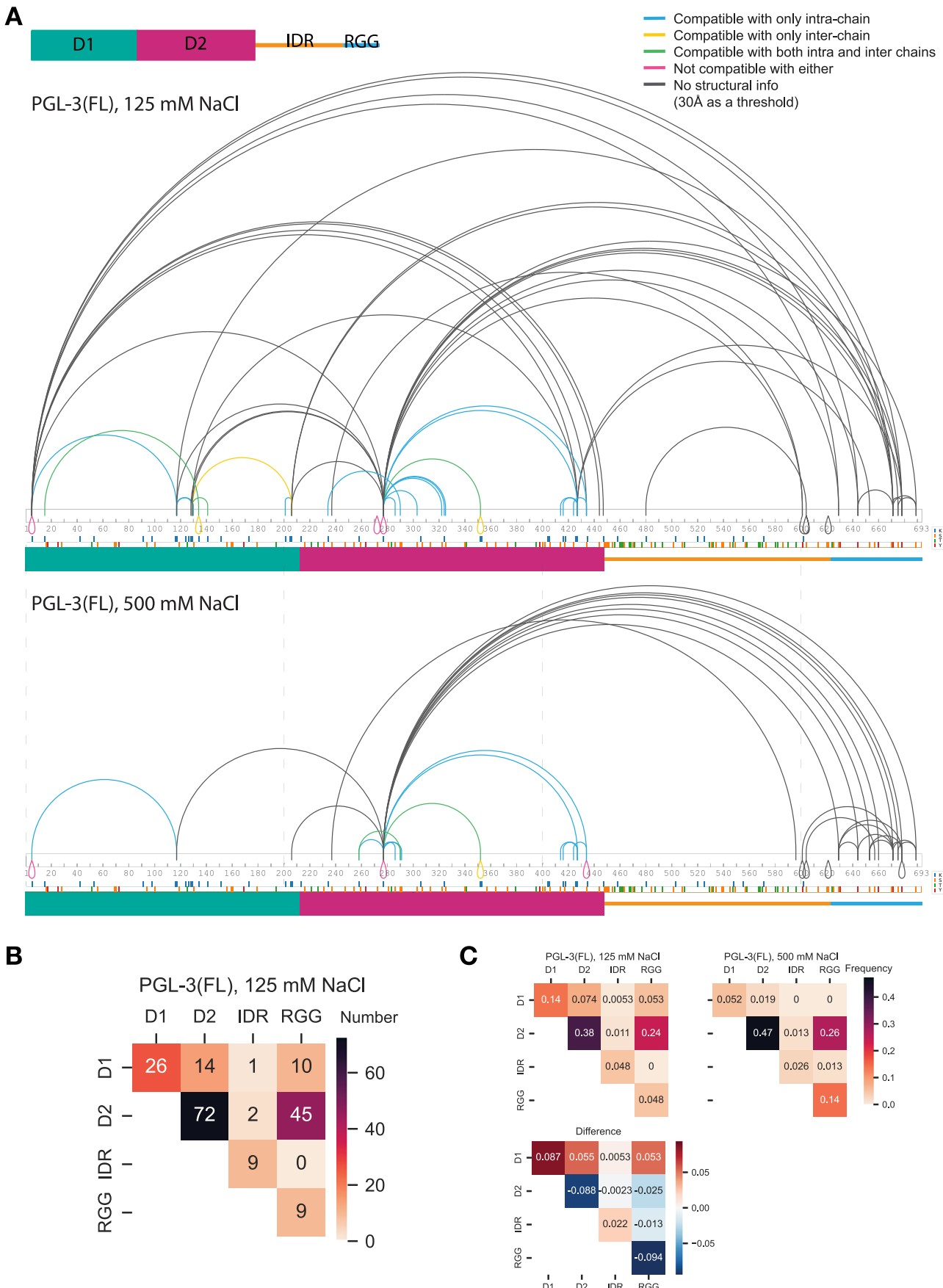

**Figure 4.    Intra- and inter-domain interactions revealed by XL-MS.**

(A) Connectograms showing all residue pairs identified by XL-MS on full-length PGL-3 in 125 mM NaCl (condensing condition) or 500 mM NaCl (non-condensing condition). Different colors indicate compatibility of each crosslink to structural models if the residue pair is intra-chain (monomer) and/or inter-chain (dimer) (Aoki et al, 2016, 2021). Blue: compatible with intra-chain only, yellow: compatible with inter-chain only, green: compatible with both, red: not compatible with either, gray: no structural information. (B) Crosstab heatmap showing the number of crosslinked peptide-spectrum matches (XSMs) whose connecting residues reside within or between the four indicated domains at the 125 mM NaCl (condensing) condition. (C) Crosstab heatmaps showing the frequency of XSMs whose connecting residues reside within or between the four indicated domains. Top two heatmaps show crosslinks obtained at 125 mM (permissive for condensation) and 500 mM NaCl (non-permissive for condensation). The crosstab titled "difference" shows values calculated by subtracting 500 mM NaCl values from 125 mM NaCl values; positive values (red colors) indicate crosslinks obtained at higher frequencies under condensing conditions.

(Methods; Appendix Fig. S6). As a control, we used free fluorescent dye, which exhibited no enrichment or depletion into D1-D2 condensates, demonstrating minimal effects of the fluorophores on partitioning (Fig. 5B,C). We found that D1-D2 condensates could enrich the RGG peptide with a partition coefficient of $1.30 \pm 0.01$. The scrambled RGG also enriched in the D1-D2 condensates (Appendix Fig. S5B), in contrast to the KGG peptide, which was recruited less efficiently, and the internal IDR which was weakly excluded (Fig. 5B,C). We conclude that D1-D2 condensates can enrich the RGG peptide, consistent with binding between D1-D2 and the RGG region in condensates.

## The RGG region enhances phase separation and increases viscosity of D1-D2 condensates in *trans*

So far, our data demonstrate that the RGG region enhances D1-D2 condensation in *cis* (Fig. 3), can be crosslinked to D1 and D2 domains (Figs. 4 and 5A) and can be enriched in D1-D2 condensates when provided in *trans* (Fig. 5B,C). We wondered if heterotypic interactions between the RGG region and D1-D2 are sufficient to modify condensate properties. To test this hypothesis, we first compared the proportion of D1-D2 in the dense phase in the presence or absence of the RGG peptide. We found that addition of the RGG peptide increased the fraction of D1-D2 in the pellet by ~15% compared to buffer control (Fig. 5D). We observed a similar effect using the scrambled RGG peptide and a weaker effect with the KGG peptide (Fig. 5A; Appendix Fig. S5C). Addition of the internal IDR did not yield a statistically significant change (Fig. 5D). We conclude that the RGG region weakly enhances phase separation of D1-D2 when provided in *trans*.

Next, we tested whether the RGG region also increases the viscosity of D1-D2 condensates. Using embedded beads, we measured the viscosity of D1-D2 condensates formed in the presence of equimolar IDR, RGG or KGG (Methods). Only the RGG increased condensate viscosity compared to buffer (by ~1.25 fold, Fig. 5E). We conclude that interactions between the RGG region and D1-D2 decrease $c_{sat}$ and increase condensate viscosity.

### D1-D2 and RNA compete for binding to the RGG region

The RGG region of PGL-3 was previously shown to bind RNA in vitro (Saha et al, 2016). To weigh the relative effects of protein-protein interactions and protein-RNA interactions, we compared the $c_{sat}$ of full-length PGL-3 and D1-D2, in the presence or absence of *nos-2* mRNA (Fig. EV5A). Full-length PGL-3 exhibited a lower $c_{sat}$ in the presence of RNA compared to the no-RNA condition. In contrast, RNA did not change the $c_{sat}$ of D1-D2, consistent with the RGG region being responsible for RNA binding (Saha et al, 2016).

Interestingly, the absolute $c_{sat}$ difference between D1-D2 and FL was larger than between FL and FL + RNA.

We used biolayer interferometry (BLI) to compare the affinity of the RGG region to D1-D2 versus RNA (Methods). We detected binding between the RGG peptide and D1, but not D2, even at higher concentrations (Fig. EV5B). The RGG peptide bound D1 and 30-nt poly(U) RNA in a concentration-dependent manner (Fig. 6A), allowing us to determine binding kinetics by fitting to a 1:1 binding model. Binding kinetics ($k_{on}$ and $k_{off}$) were ~20 and 10-fold faster, respectively, for RGG:D1 compared to RGG:RNA, resulting in a ~twofold lower $K_D$ (higher affinity) for RGG:D1 compared to RGG:RNA. The slower dynamics of RGG:RNA interactions are consistent with the previously reported increased immobile FRAP fraction of full-length PGL-3 condensates supplemented with mRNA (Saha et al, 2016). These results suggest D1 effectively competes against RNA for binding to the RGG region.

To assess the relative interaction strengths of RGG:D1-D2 and RGG:RNA interactions in the context of phase separation, we compared partitioning of the RGG peptide into D1-D2 condensates in the presence or absence of in vitro transcribed *nos-2* mRNA. When tested at a physiological stoichiometry of PGL-3 and RNA calculated based on values reported previously (Saha et al, 2016) (4 μM PGL-3 and 300 ng/μl *nos-2* RNA; Methods), the RGG region was still enriched in D1-D2 condensates, although at a lower partition coefficient compared to the no-RNA condition (Fig. 6B,C). A higher RNA concentration of 500 ng/μl decreased partitioning of the RGG peptide into D1-D2 condensates, consistent with RNA competing with D1-D2 for RGG binding. The RGG peptide also formed aggregates with RNA in the presence or absence of D1-D2 (Figs. 6B and EV5C). We conclude that RNA and D1-D2 compete for RGG binding and that D1-D2:RGG interactions are maintained in the presence of physiological levels of RNA, implying that these may be favored when the RGG region is covalently linked to D1-D2, as in full-length PGL-3.

## Discussion

In this study, we have characterized the molecular mechanism of phase separation by the P granule protein PGL-3. Contrary to computational predictions, we find that the folded domains of PGL-3 (D1 and D2) are necessary and sufficient for phase separation. Crosslinking and mass photometry experiments suggest that these domains confer multivalency to PGL-3, allowing it to form oligomers larger than dimers in solution and in condensates. D1-D2 also interacts with the C-terminal RGG region, and these heterotypic (e.g., D1:RGG) interactions enhance phase separation

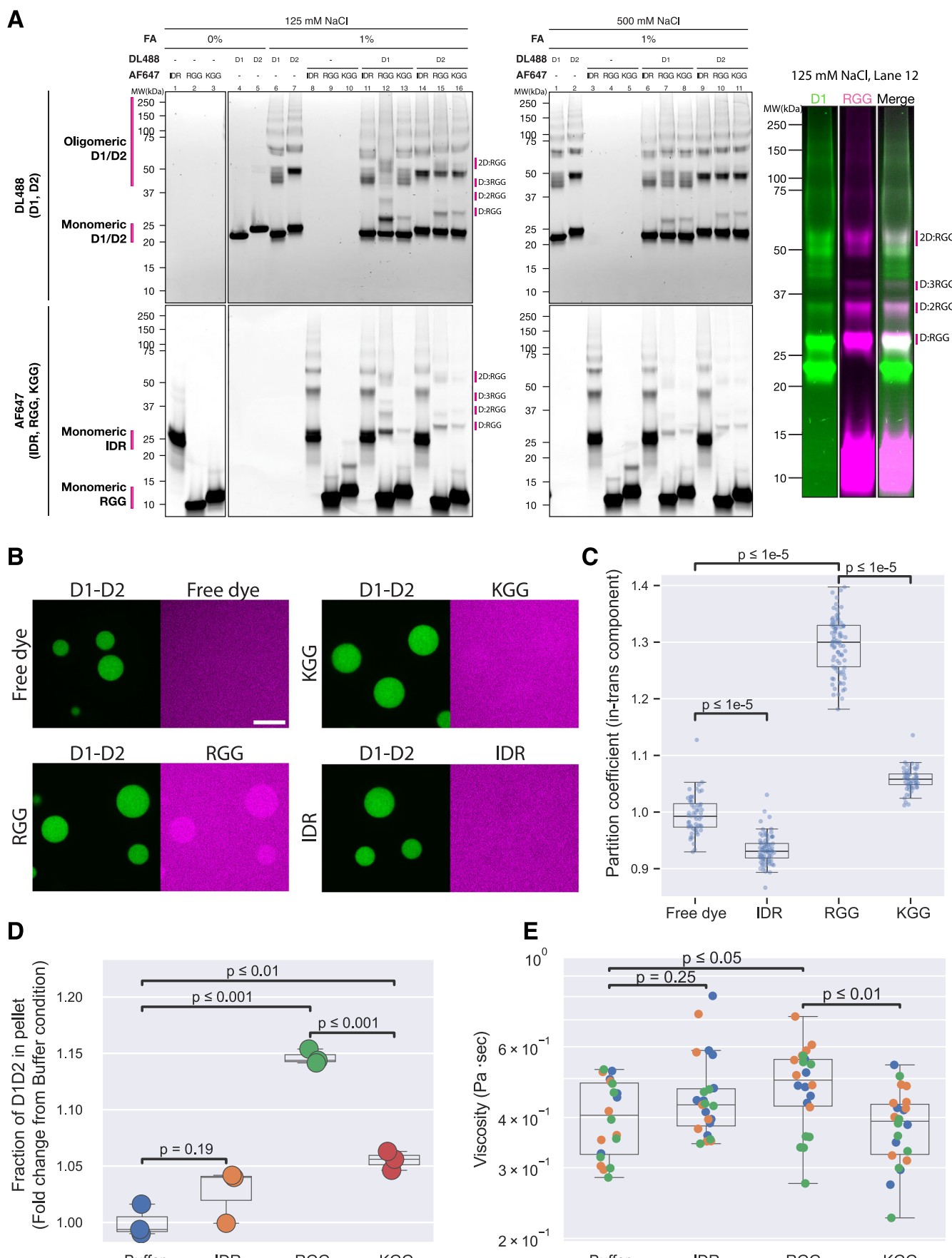

**Figure 5. The RGG region binds to D1 and D2 in trans and modulates D1-D2 phase separation.**

(A) Fluorescence images of SDS-PAGE gels showing crosslinked complexes of PGL-3 fragments. Top and bottom images were obtained from the same gel, scanned for the two different fluorophores. Top: D1 and D2 labeled with DyLight 488. Bottom: IDR, RGG and KGG labeled with Alexa Fluor 647. Color image is color version of lane 12 from the 125 mM NaCl condition. (B) Micrographs comparing partitioning of the IDR, RGG and KGG (magenta) to D1-D2 condensates (green). Results are quantified in C. Scale bar = 10 μm. (C) Plot showing the partition coefficients of component supplied in trans to D1-D2 condensates. Each dot represents a value from a single condensate. In the overlaying box plots, lines inside boxes show median (Q2), box bounds are quartiles (Q1 and Q3), and whiskers show the range of data excluding outliers identified by the Turkey method. $P$ values calculated by Wilcoxon rank-sum test with Benjamini–Hochberg adjustment for multiple comparisons. Technical replicates. $N > 56$ condensates per condition. $P$ values $= 4.669 \times 10^{-19}$ (FreeDye vs. IDR), $5.789 \times 10^{-24}$ (FreeDye vs. RGG) and $2.244 \times 10^{-25}$ (RGG vs. KGG). (D) Plot showing the fraction of D1-D2 protein in pellet over total protein, normalized to the buffer control, comparing different PGL-3 domains added in trans. In the overlaying box plots, lines inside boxes show median (Q2), box bounds are quartiles (Q1 and Q3), and whiskers show the range of data excluding outliers identified by the Turkey method. Briefly, D1-D2 (trace-labeled with fluorophore) condensation was induced in buffer alone or buffer supplemented with the peptide indicated on the x-axis (equimolar concentration). The reactions were incubated and centrifuged to separate the dense phase (pellet fraction) from the dilute phase (supernatant). The amounts of D1-D2 in each fraction were measured by fluorescence intensity following SDS-PAGE to resolve D1-D2 from the added components. $P$-values calculated by Wilcoxon rank-sum test with Benjamini–Hochberg adjustment for multiple comparisons. Technical replicates. $N = 3$. $P$ values $= 0.185$ (Buffer vs. IDR), $7.195 \times 10^{-4}$, (Buffer vs. RGG), $1.747 \times 10^{-4}$ (RGG vs. KGG), and $8.166 \times 10^{-3}$ (Buffer vs. KGG). (E) Plot showing the estimated viscosity of D1-D2 condensates assembled in buffer alone or buffer supplemented with the peptide indicated on the x-axis. In the overlaying box plots, lines inside boxes show median (Q2), box bounds are quartiles (Q1 and Q3), and whiskers show the range of data excluding outliers identified by the Turkey method. Viscosity measurement was done by passive rheology (Methods). $P$ values calculated by Wilcoxon rank-sum test with Benjamini–Hochberg adjustment for multiple comparisons. Technical replicates. $N > 20$ condensates. $P$ values: $0.2474$ (Buffer vs. IDR), $0.01407$ (Buffer vs. RGG), and $1.355 \times 10^{-3}$ (RGG vs. KGG). Source data are available online for this figure.

and increase the viscosity of PGL-3 condensates. In contrast, the internal IDR was fully soluble under the conditions tested and is neither necessary nor sufficient for phase separation. Based on these observations, we propose the following working model for PGL-3 phase separation (Fig. 7). D1 and D2 both form oligomers and provide the requisite multivalency for condensation. The RGG region further enhances phase separation by interacting in trans with D1-D2. The internal IDR contributes indirectly to phase separation by serving as a flexible linker between D1-D2 and RGG. We discuss aspects of this model in the following sections.

## PGL-3 condensation is driven primarily by oligomerization of the folded D1 and D2 domains

Our systematic screening of PGL-3 derivatives revealed that the first 447 amino acids of PGL-3, which span the folded domains D1 and D2, are necessary and sufficient for phase separation. Analyses of the D1 and D2 domains by X-ray crystallography revealed that both can form homodimers (Aoki et al, 2016, 2021). Using XL-MS, we identified seven unique homodimer crosslinks in D1 and D2 compatible with the published homodimer structures (Aoki et al, 2016, 2021), suggesting that the dimerization models observed in crystals are relevant in solution and in liquid condensates. We also identified four additional unique homodimer crosslinks that were not compatible with the published structures, suggesting that D1 and D2 also dimerize using interfaces not seen in the X-ray structures. Consistent with these findings, we detected homo-oligomers larger than dimers for both D1 and D2 using FA crosslinking in solution. We conclude that D1 and D2 are oligomerization domains that contain at least two homotypic binding interfaces that can be co-occupied on the same molecule. We suggest that D1 and D2 domains provide the multivalency required to create higher-order networks of PGL-3 molecules and are the primary drivers of PGL-3 phase separation.

## PGL-3 condensation is enhanced by interactions between D1-D2 and the RGG region

Several lines of evidence indicate that D1-D2 also interact with the RGG region. By XL-MS, we detected 10 and 45 spectra corresponding to crosslinks between the RGG region and D1 and the RGG region and D2, respectively, under phase-separating conditions. The D1 domain, which is furthest away in sequence from the RGG region, interacts with the RGG region only under phase-separating conditions in XL-MS experiments. We observed hetero-oligomers of various stoichiometries in crosslinked mixtures of RGG peptide and D1 or D2 domains. Furthermore, we detected binding between the D1 domain and the RGG region by biolayer interferometry (BLI). Consistent with these interactions contributing to PS, we observed that, when provided in trans, the RGG peptide enriches in D1-D2 condensates and (modestly) lowers $c_{sat}$, and when covalently attached to D1-D2 (cis effect), the RGG region increases condensate viscosity and lowers $c_{sat}$. We conclude that the RGG region interacts with the structured domains of PGL-3 in a manner that enhances PS.

We propose two related models for how the RGG region tunes the phase separation properties of D1-D2. In the first model, binding between the RGG region and D1-D2 simply increases total valency, which augments the PGL-3 network, lowering $c_{sat}$ and increasing condensate viscosity. In a second model, the RGG region competes for one of the binding surfaces used by D1-D2 to oligomerize, which alters the landscape and kinetics of the network. In support of the second model, residues in D1 and D2 (K5 and K277) were detected by XL-MS in both homodimer crosslinks and heterotypic crosslinks involving the RGG region (Fig. 4A), raising the possibility that the same interface(s) that mediate D1 and D2 homodimer interactions also mediate D1-D2:RGG interactions. One possibility is that certain homotypic binding interfaces in D1-D2 lead to assemblies that are less capable of forming a dynamic higher-order network and therefore tend to terminate at soluble oligomers. Competition for these sites by the RGG region could theoretically enhance condensation if oligomers involving the RGG region were less soluble and/or more prone to forming higher-order networks. A scrambled RGG peptide behaved similarly to the native RGG peptide in crosslinking to D1 and D2, enhancing the D1-D2 condensation and recruitment to D1-D2 condensates, suggesting that the specific sequence of the RGG region is not essential, consistent with variability in the numbers of RGG motifs between PGL-1 and PGL-3. We also note

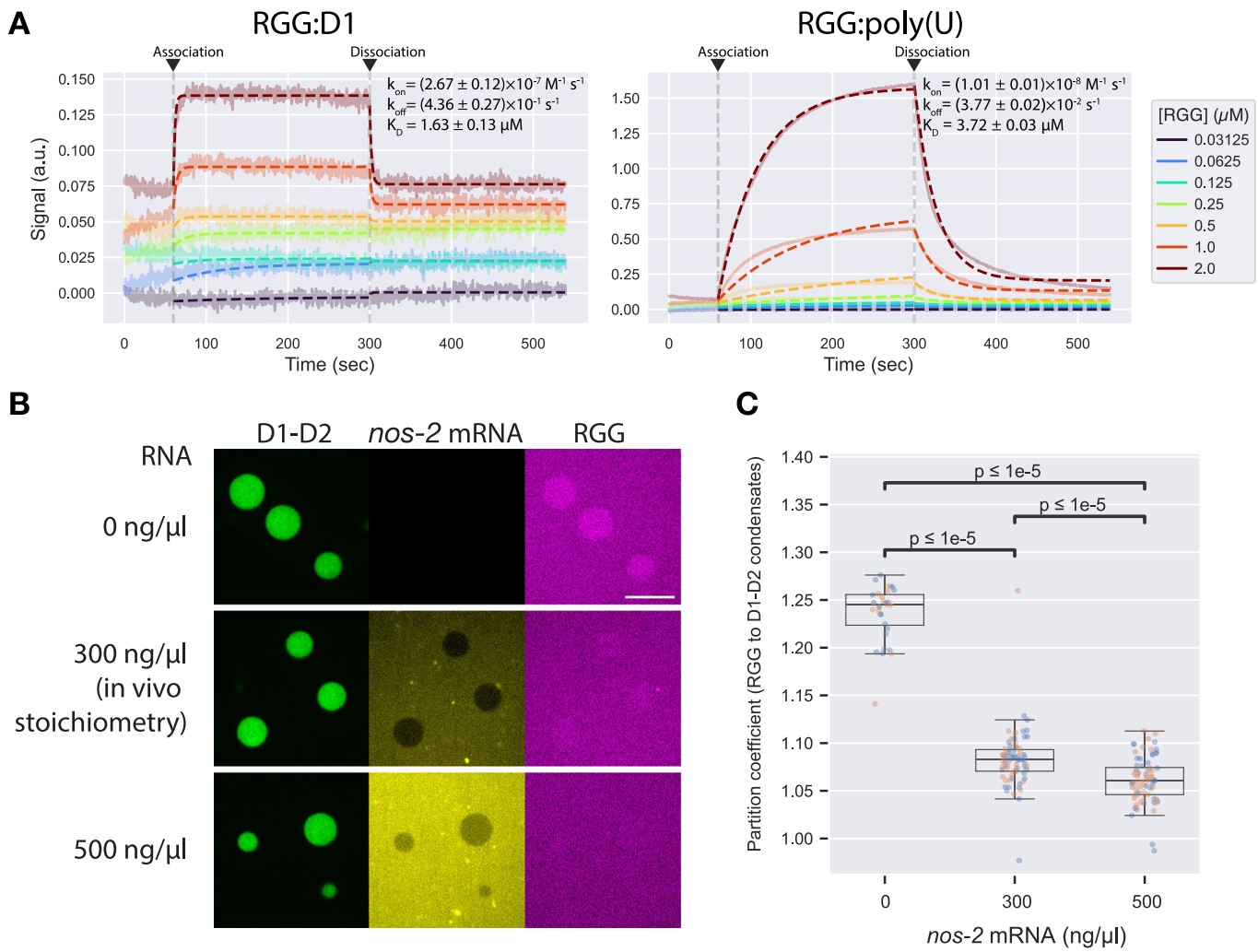

**Figure 6. D1-D2 competes with RNA for binding to RGG region.**

(A) BLI sensorgrams showing the RGG peptide binding to PGL-3(D1) or poly(U) RNA. The experiments were performed with D1 or RNA on the sensor tips and the RGG peptide in solution. Opaque solid lines are experimental data, and dotted lines are model fits. (B) Representative fluorescence micrographs showing distribution of D1-D2 (green), *nos-2* mRNA (yellow) and the RGG peptide (magenta) upon D1-D2 condensation, at 4 μM D1-D2 and 4 μM RGG with varying concentrations of RNA and 125 mM NaCl. Top row, middle pane: in the absence of RNA, only low-level noise signals are observed, which do not show because fluorescence intensities are levelled across conditions, resulting in an image devoid of signal. (C) Plot showing the partition coefficients of the RGG peptide to D1-D2 condensates at varying concentrations of RNA. In the overlaying box plots, lines inside boxes show median (Q2), box bounds are quartiles (Q1 and Q3), and whiskers show the range of data excluding outliers identified by the Turkey method. *P* values calculated by the Wilcoxon rank-sum test with Benjamini–Hochberg adjustment for multiple comparisons. Technical replicates. N > 79 condensates per condition. *P* values = $2.248 \times 10^{-15}$ (0 ng/μl vs. 300 n/μl), $4.404 \times 10^{-7}$ (300 ng/μl vs. 500 ng/μl), and $2.884 \times 10^{-16}$ (0 ng/μl vs. 500 ng/μl).). Source data are available online for this figure.

that the effect of the RGG region on condensation was considerably higher in *cis* than in *trans*. A likely possibility is that covalent linkage of the RGG to D1-D2 increases the effective $k_{on}$ between RGG and D1-D2. We cannot exclude, however, the possibility that the RGG region also enhances phase separation by forming RGG homo-oligomers as has been reported for other RGG-rich regions (Poornima et al, 2019; Shaw et al, 2010). Crosslinks within the RGG region were recovered in the XL-MS data, but were not more prevalent under condensing conditions. We attempted to generate point mutants that specifically blocked D1-D2:RGG interactions, but unfortunately, these also interfered with phase separation of D1-D2 alone, potentially indicative of the

dual use of binding interfaces as discussed above. The observation that the RGG region impacts D1-D2 condensation in *trans* is strong evidence that the RGG region does not function solely through RGG-RGG interactions.

## Does RNA tune PGL-3 condensation in vivo?

The RGG region of PGL-3 has also been proposed to enhance phase separation by binding to RNA (Saha et al, 2016). Our BLI and partitioning competition assays indicate that D1-D2 effectively competes with RNA for binding to the RGG region, and that both binding interactions can be detected at stoichiometries of

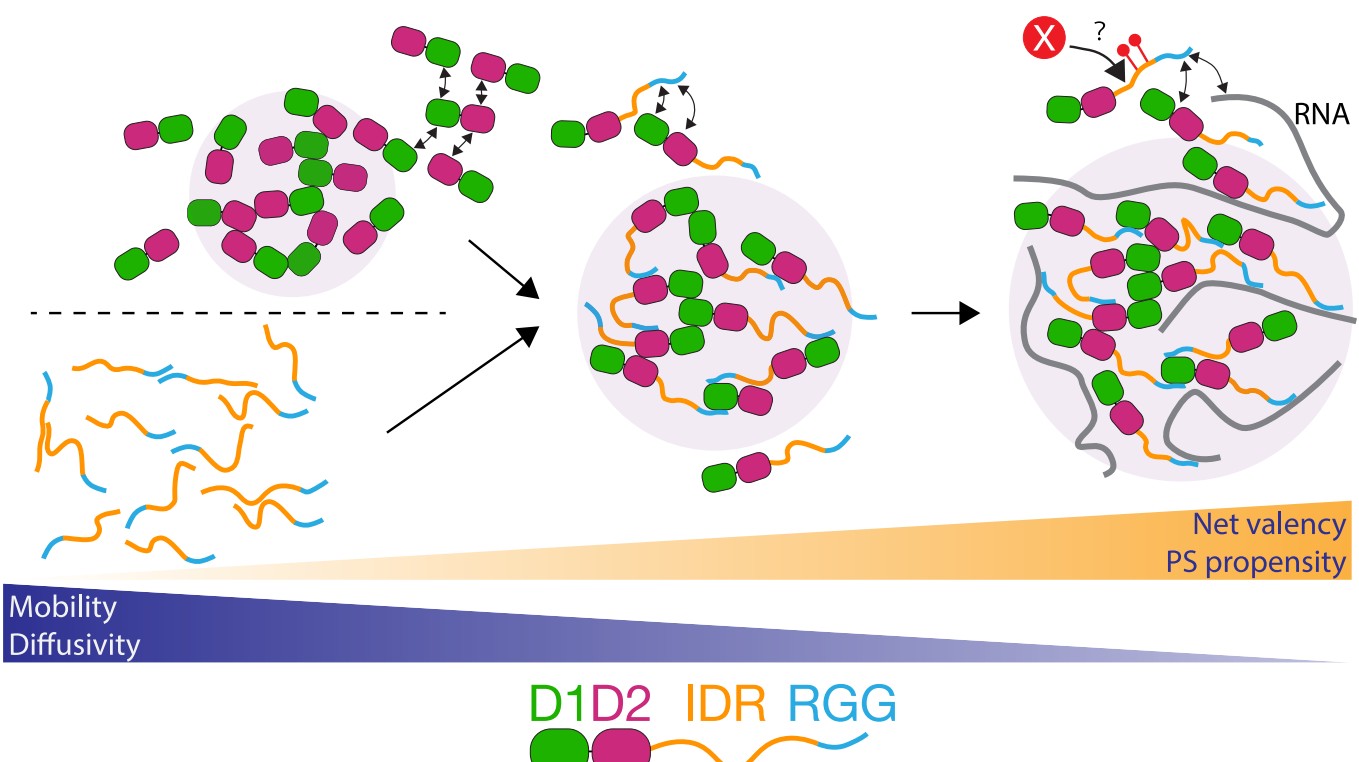

**Figure 7. Proposed model for PGL-3 phase separation.**

Schematics showing the interactions that drive PGL-3 phase separation (PS). The schematic below summarizes the roles of each domain. Top left panel: D1 and D2 domains are oligomerizing domains and provide the foundation of PGL-3 PS. Bottom left panel: IDR-RGG have limited self-associative interactions and do not undergo PS. Middle panel: IDR-RGG region enhances PS by promoting binding interactions between D1-D2 and the RGG region. Right panel: RNA further enhances PS by binding to the RGG region. The internal IDR may contain sites for binding to other proteins or for post-translational (PTMs) modifications.

PGL-3 and RNA that mimic those in vivo. Whether PGL-3 interacts with RNA in vivo is not yet known. Compared to other RNA-binding proteins present in *C. elegans* embryos (e.g., MEX-5 and MEG-3), PGL-3 has a lower binding affinity for RNA (Schmidt et al, 2021; Saha et al, 2016; Lewis et al, 2025). Competition between PGL-3 and MEX-5 for RNA binding has been invoked as a mechanism to bias PGL-3 PS to the posterior cytoplasm of zygotes, where MEX-5 concentration is lowest (Lewis et al, 2025; Saha et al, 2016). Studies from our laboratory, however, have provided evidence for another mechanism driving P granule polarization in embryos that does not require PGL binding to RNA. In this model, MEX-5 competes for RNA with MEG-3, driving MEG-3 to the posterior cytoplasm, where MEG-3 stabilizes P granules by forming interfacial clusters on PGL droplets (Pickering effect) (Smith et al, 2016; Folkmann et al, 2021). Genetic analyses have demonstrated that mRNAs are recruited to P granules by MEG-3, independent of PGL-1 or PGL-3. Furthermore, iCLIP experiments comparing MEG-3 and PGL-1 identified 657 transcripts bound to MEG-3 and only 18 transcripts bound to PGL-1 (Lee et al, 2020). Together, these observations raise the possibility that PGL:RNA binding is limited in vivo,

leaving the RGG motifs available for binding to D1-D2. In our hands, the PGL-3 $c_{sat}$ in the absence of RNA in vitro is 0.25 and 0.7 µM in 125 mM NaCl and 150 mM NaCl, respectively, within the range of concentration estimates for PGL-3 (0.68–0.96 µM) and its co-expressed paralogue PGL-1 (1.3 µM) in cells (Saha et al, 2016; Fritsch et al, 2021). PGL proteins, therefore, may be present at sufficiently high levels in vivo to phase separate without the help of RNA. Additional in vivo studies will be required to determine whether RNA influences PGL phase separation in cells.

## The internal IDR of PGL-3 contributes to phase separation indirectly by acting as a linker between the D1-D2 and RGG region

Our experimental findings indicate that the central IDR in PGL-3 is neither necessary nor sufficient for phase separation. XL-MS and FA crosslinking revealed few interactions between the internal IDR and D1-D2, the IDR was slightly depleted from D1-D2 condensates, and D1-D2-IDR underwent phase separation at a higher $c_{sat}$ than D1-D2. These data suggest that auto-

inhibitory mechanisms involving binding between D1-D2 and the internal IDR are unlikely. Rather, we suggest that the internal IDR could be acting as a passive solubilizer. Consistent with this view, the IDR remained soluble in all the conditions we tested, and FINCHES (Ginell et al, 2025) predicts that the internal IDR is mostly repulsive to itself and contains only small regions of weakly attractive patches (Appendix Fig. S7).

Deletion of the internal IDR in PGL-3 led to a minor increase in $c_{sat}$, which could not be rescued by insertion of a short 3x(GGGGS) linker, raising the possibility that the IDR could be making a small contribution to phase separation by sequence-dependent associative effects and/or length-dependent linker effects. Consistent with the latter, the IDRs of PGL-1 and PGL-3 are both relatively long (223 and 175 amino acids) despite low overall sequence identity (43%) compared to D1-D2 (69%) and the RGG region (61%) (Appendix Fig. S1A,B). The IDR could also provide sites for protein modifications that could tune phase separation. Phosphorylation of PGL-1 and PGL-3 by the mTOR homolog LET-363 was reported to enhance phase separation in vitro (Zhang et al, 2018). The same study showed that three out of five phosphorylation sites reside in the internal IDR of PGL-1. In PGL-3, phosphoproteomics studies identified eight phosphorylation sites, all located in the IDR (Lin et al, 2021; Edifizi et al, 2017; Huang et al, 2018; Zielinska et al, 2009). It will be interesting to explore the impact of these post-translational modifications on phase separation.

## Limitations of this study

Our study explored the molecular basis of PGL-3 phase separation using in vitro assays. Although our findings are consistent with observations that demonstrated the importance of D1-D2 for PGL condensation in cells, we did not investigate directly whether the interactions we uncovered in vitro also occur in vivo. A major challenge is that, unlike our assays, which examined PGL-3 condensation in isolation, PGL-3 condensation in vivo occurs in the presence of dozens of other P granule proteins, many of which also contribute to P granule assembly (Thomas et al, 2025). In fact, a preprint reported that introducing other P granule proteins can modify the dynamics of PGL-3 co-condensates assembled in vitro (preprint: Jelenic et al, 2024). We cannot exclude the possibility that in vivo, PGL-3 phase separation involves interactions with additional proteins that bind to D1-D2, the internal IDR, and/or the RGG region. Another challenge is that our buffer conditions do not accurately mimic the properties and crowding of the *C. elegans* cytoplasm. Although we could observe D1-D2 oligomers at relatively low concentrations (50-100 nM), D1-D2 condensation required higher concentrations (~1 µM), near the range estimated for PGL-1/3 concentration in embryos (1.3 µM(Saha et al, 2016) for PGL-1 and 0.68–0.96 µM (Saha et al, 2016; Fritsch et al, 2021) for PGL-3). Finally, as mentioned above, PGL proteins are modified in vivo and these modifications could alter their phase separation properties (Li et al, 2013; Zhang et al, 2018). Nonetheless, the reductionist approach used in this study demonstrates that 1. folded domains can provide sufficient multivalency for phase separation and 2. RGG repeats can augment phase separation independent of RNA binding.

# Methods

### Reagents and tools table

| Reagent/resource | Reference or source | Identifier or catalog number |
|---|---|---|
| **Recombinant DNA** | | |
| pRK015 6xHis::MBP::6xHis::TEV::PGL-3(FL) | This study | Addgene 250134 |
| pRK049 6xHis::MBP::6xHis::TEV:PGL-3(D1) | This study | Addgene 250135 |
| pRK050 6xHis::MBP::6xHis::TEV::PGL-3(D2) | This study | Addgene 250136 |
| pRK011 MBP::6xHis::TEV::PGL-3(IDR) | This study | Addgene 250137 |
| pRK036 6xHis::MBP::6xHis::TEV::PGL-3(D1-D2) | This study | Addgene 250138 |
| pRK042 6xHis::MBP::6xHis::TEV::PGL-3(D1-D2-IDR) | This study | Addgene 250139 |
| pRK082 6xHis::MBP::6xHis::TEV::PGL-3(D1-D2-RGG) | This study | Addgene 250140 |
| pRK084 6xHis::MBP::6xHis::TEV::PGL-3(D1-D2-3(GGGGS)-RGG) | This study | Addgene 250141 |
| pAAP35 MBP::6xHis::TEV::PGL-3(IDR-RGG) | This study | Addgene 250142 |
| pAAP36 MBP::6xHis::TEV::PGL-3(D1-IDR-RGG) | This study | Addgene 250143 |
| pAAP37 MBP::6xHis::TEV::PGL-3(D2-IDR-RGG) | This study | Addgene 250144 |
| pRK047 6xHis::MBP::6xHis::TEV::PGL-3(FL) (R123E, K126E, K129E) | This study | Addgene 250145 |
| **Oligonucleotides and other sequence-based reagents** | | |
| PGL-3(RGG) | GenScript | NA |
| PGL-3(KGG) | GenScript | NA |
| PGL-3(scrambled RGG) | GenScript | NA |
| **Chemicals, enzymes and other reagents** | | |
| HEPES, 1.0 M buffer, pH 7.5 | Thermo Fisher | J60712.AP |
| DyLight™ 488-NHS Ester | Thermo Fisher | 46402 |
| Alexa Fluor™ 647 NHS Ester (Succinimidyl Ester) | Invitrogen | A37573 |
| ChromaTide™ Alexa Fluor™ 488-5-UTP | Invitrogen | C11403 |
| ChromaTide™ Alexa Fluor™ 546-14-UTP) | Invitrogen | Discontinued |
| mMESSAGE mMACHINE™ T7 Transcription Kit | Invitrogen | AM1344 |
| FluoSpheres™ Size Kit | Invitrogen | F8888 |
| **Software** | | |
| Fiji | Open-source | http://fiji.sc |
| Python | Open-source | https://www.python.org |
| TrackMate | Ershov et al, (2022) | |
| **Other** | | |
| AXIO Observer | Zeiss | |

| Reagent/resource | Reference or source | Identifier or catalog number |
|---|---|---|
| CSU-W1 SoRA | Yokogawa | |
| UltiMate3000 UHPLC | Thermo Fisher | |
| Q-Exactive HF-X Orbitrap | Thermo Fisher | |
| TwoMP | Refeyn | |

## Computational predictions and calculation of physicochemical parameters

Disorder predictions were performed on the MetaDisorder server (Kozlowski and Bujnicki, 2012) and metapredict V3 (preprint: Lotthammer et al, 2024), and the charge distribution was calculated on EMBOSS-charge (Rice et al, 2000) with the window length = 9. Phase separation prediction was performed on their respective servers for DeePhase (Saar et al, 2021), FuzDrop (Hardenberg et al, 2020), PSPHunter (Sun et al, 2024) and MolPhase (Liang et al, 2024). For PSPire, the prediction was run locally following the authors' instructions (Hou et al, 2024). Molecular weights, pIs and charges were calculated on Prot pi (https://www.protpi.ch).

## Protein preparation

The RGG peptide and its derivatives were synthesized chemically. Other PGL-3 derivatives were expressed as fusions with an N-terminal His-MBP purification tag in *E. coli* Rosseta2 (DE3) cells (Sigma). The fusions were purified, and the His-MBP tag was removed through four purification steps (amylose affinity, reverse nickel affinity, heparin-affinity or anion exchange, size exclusion) following published protocols with modifications (Putnam and Seydoux, 2021) on an ÄKTA pure FPLC or using loose resins. For fluorophore labeling, amine-reactive Alexa Fluor 647 NHS ester (Invitrogen) and DyLight 488-NHS-ester (Thermo Fisher) were used.

## Molecular cloning

Plasmids used in this study are listed in Appendix Table S1. Plasmids were prepared using standard molecular cloning procedures in *E. coli* DH5α (Sigma) using the pMAL-c2X vector backbone. PGL-3 sequences were fused at their N-termini with either a 6xHis-MBP-6xHis-TEV or a MBP-6xHis-TEV purification tag, with or without a linker between the TEV recognition sequence and PGL-3.

## Protein expression and purification

### Protein expression

Transformed *E. coli* Rosseta2 (DE3) cells were grown in lysogeny broth (LB) overnight, then subcultured in terrific broth supplemented with 75 µg /ml ampicillin at 37 °C, 250 rpm until reaching an OD600 of ~1.0. The culture was cooled down at 16 °C for 10 min, and 1 mM isopropyl β-D-1-thiogalactopyranoside(IPTG) was added to induce protein expression overnight at 16 °C with 250 rpm shaking. Cells were collected by centrifugation, and the pellets

were either directly used for purification or frozen and kept at −80 °C until further use. Bacterial pellets were sonicated in lysis buffer [10 µg/ml RNaseA(QIAGEN), 1 tablet/10 ml cOmplete™ Mini(EDTA-free) (Roche), 250 mM NaCl, 0.4 M L-arginine, 10% (v/v) glycerol, 1 mM DTT, 12.5 mM HEPES, pH 7.5].

### Amylose affinity chromatography
Lysed cells were cleared by centrifugation and filtration, and the lysate was bound to loose amylose resin (NEB) in a gravity column or MBPTrap HP 5 ml column (Cytiva). The resin was washed with MBPNaCl buffer [20% (v/v) glycerol, 500 mM NaCl, 50 mM HEPES, pH 7.5, 1 mM DTT] and eluted with MBPNaCl buffer supplemented with 20 mM maltose.

### TEV cleavage and reverse nickel affinity chromatography
In-house purified 6xHis-TEV protease was added to protein elution fractions to cleave the tag at 16 °C overnight. For home-made TEV protease, pET29b-10xHis-Super TEV was acquired from Addgene; the 10xHis tag was reduced to 6xHis for convenience, and purified following the published protocol (Lau et al, 2023). The digested purification tag and undigested fractions were removed by reverse His-tag affinity chromatography on a HisTrap HP 5 ml column (Cytiva).

### Heparin-affinity or anion exchange chromatography
PGL-3 fragments were purified further using a HiTrap Heparin HP 5 ml column (Cytiva) (for constructs containing the RGG region) or a HiTrap Q HP 5 ml column (Cytiva) (for constructs lacking the RGG region). The flow-through from reverse nickel affinity chromatography was collected and diluted fivefold to reach 100 mM NaCl with dilution buffer [40%(v/v) glycerol, 0.1% NP-40, 25 mM HEPES pH 7.5], loaded onto a column, washed with 20%(v/v) glycerol, 100 mM NaCl, 25 mM HEPES, pH 7.5, and eluted by gradient elution from 150 to 1000 mM NaCl.

### Size exclusion chromatography and storage
Protein fractions were run on HiPrep Sephacryl S-200 (Cytiva) in native buffer [20%(v/v) glycerol, 500 mM NaCl, 25 mM HEPES, pH 7.5, 0.5 mM TCEP] and dialyzed against native buffer. Proteins were concentrated using appropriately-sized Ultra Centrifugal Filters (Amicon), filtered with a 0.2 µm PES filter (MDI), aliquoted, flash frozen in liquid nitrogen, and stored at −80 °C. Protein concentration of the final stock was determined by absorbance at 280 nm, and protein purity was assessed by SDS-PAGE followed by Coomassie Brilliant Blue G250 staining (Fig. EV1A).

## Preparation of chemically synthesized peptides

RGG, KGG and scrambled RGG peptides were custom-synthesized by GenScript. In the KGG peptide, all arginine residues are replaced with Lys (Fig. EV4C). The scrambled RGG peptide was designed to scramble the sequence of the native RGG peptide and disrupt all RGG triplets. However, it contains six RG doublets at non-native positions, whereas the native RGG peptide contains six RGG triplets and six RG doublets (Fig. EV4C). We used this scrambled RGG peptide that contains RG doublets after repeated failures to synthesize peptides disrupting all RGG and RG motifs. The lyophilized peptides were suspended and dialyzed against the native buffer to remove residual salt from synthesis. Peptides were filtered using a 0.2-µm PES filter (MDI) and flash frozen in liquid

nitrogen and stored at −80 °C. Protein concentration of the final stock was determined by absorbance at 280 nm.

## Protein labeling with a fluorophore

Proteins were diluted to 25 μM and mixed with ~50 μM Alexa Fluor 647 NHS ester(Invitrogen) or DyLight 488-NHS-ester(Thermo Fisher) in native buffer, and reacted for 1–4 h in the dark, with the exception of the KGG peptide, which was reacted in 20%(v/v) glycerol, 500 mM NaCl, 50 mM phosphate buffer pH 6.5 at 4 °C overnight to preferentially label the N-terminal amine group. Free dyes were removed by running the solution in Zeba™ Spin Desalting Columns, 7 K MWCO (Thermo Fisher), three times. Labeled proteins were filtered using a 0.2-μm PES filter (MDI), aliquoted, flash frozen in liquid nitrogen, and stored at −80 °C. Protein and dye concentrations were measured using Nanodrop One. The dye to protein ratio never exceeded 1.0, indicating less than one fluorophore per molecule on average. Note that RGG and KGG peptides were run through the desalting columns despite their smaller molecular weight; although the recovery was lower, the final yield was sufficient for our purposes.

## RNA synthesis

nos-2 mRNA was synthesized using mMESSAGE mMACHINE™ T7 Transcription Kit (Invitrogen) following the manufacturer's recommendation as described (Lee et al, 2020). The DNA template was PCR amplified from a plasmid containing the nos-2 cDNA sequence. ChromaTide™ Alexa Fluor™ 488-5-UTP or 546-14-UTP (Invitrogen) was included at 1/20 of the reaction volume to trace-label fluorescently. LiCl precipitated RNAs were dried, resuspended in nuclease-free water, and the integrity was verified by denaturing agarose gel electrophoresis.

## Microscopy

Microscopy was carried out at the ambient temperature of ~19 °C using an AXIO Observer (Zeiss) with a CSU-W1 SoRA spinning disk confocal system (Yokogawa) and an iXon Life 888 EMCCD camera (Andor) using SlideBook software (Intelligent Imaging Innovations). The SoRaD1 filter set was used for all imaging, including fluorescent and DIC. Unless otherwise noted, images were taken using a 100x objective lens at 100 ms exposure. For viscosity measurements, the time-lapse images were acquired using a 100x objective with a 2.8x relay lens at 3 ms exposure and 150 ms interval.

## Condensation assay and phase diagrams

To prepare imaging sides, coverslips were sonicated in 50% Isopropyl alcohol, blow-dried and stored avoiding dust at room temperature. Coverslips were mounted on Attofluor Cell Chamber (Invitrogen) or sealed onto LabTek Chamber Slide (Thermo Fisher), and 3 mm diameter × 1 mm depth CultureWell silicone gasket (Grace Bio-labs) was applied to create imaging wells. Buffers, proteins and RNA were mixed directly in an imaging well to induced condensation at the desired protein and NaCl concentrations. When indicated, fluorophore-labeled proteins were mixed to aid imaging, but only up to 1% of the protein population were labelled to minimize artefacts. To create phase diagrams,

fluorophore-labelled PGL-3 solutions were scored by microscopy for visible droplets five minutes after initial mixing. Crowding agents were not used since they are not needed for PGL-3 PS, as shown by previous studies (Saha et al, 2016; Schmidt et al, 2021).

## Turbidity measurements

Condensation of unlabeled PGL-3 was induced by dilution to varying concentrations of PGL-3 and NaCl in 25 mM HEPES, pH 7.5. The solution was quickly mixed by pipetting up and down before mounting 2 μl of the NanoDrop One (Thermo Fisher) spectrometer. Turbidity was measured via absorbance at 330 nm (Appendix Fig. S2A).

## $C_{sat}$ estimation by BCA assay

Unlabeled PGL-3 were diluted to induce condensation in 20 μl reactions. Following the incubation at the room temperature (18–19 °C) for 1 h, the solution was centrifuged at 21,000×g for 20 min at 19 °C using a temperature-controlled centrifuge (Appendix Fig. S2C, Left). For $c_{sat}$ estimation with RNA, in vitro transcribed nos-2 mRNA was added at in vivo stoichiometry quantified in a previous study (Saha et al, 2016). About 18 μl from the supernatant was directly used to determine the protein concentration using Pierce™ BCA Protein Assay Kits (Thermo Fisher) according to the manufacturer's description. The reported values are mean ± 95% CI. The data were collected from >2 days with >2 independent condensation reactions per day.

## Condensate size distribution

The condensation of full-length PGL-3 or D1-D2 was induced at 5 μM PGL-3, 125 mM NaCl. Following 5 min of incubation, microscopy was performed to capture condensates floating in solution. Using custom scripts, the sizes of condensates were determined.

## Viscosity measurements

### Experiment

Viscosity was estimated by recording the diffusion of 0.2 μm diameter FluoSphere Carboxylate-Modified Microspheres (Invitrogen) in PGL-3 condensates as described in (Elbaum-Garfinkle et al, 2015) (Fig. 2C, bottom). PGL-3 condensation was induced in a buffer solution containing microspheres in 125 mM NaCl, 25 mM HEPES, pH 7.5. Condensates were allowed to settle onto the coverslip surface for 5 min, and all the image acquisition was finished within 30 min from the point of condensation. To minimize the confounding effects from interfaces, the confocal volume was placed at least 0.8 μm above and below the coverslip-condensate interface and the top of the condensate-water interface, respectively. Trace-labelled PGL-3 were used to observe the position and interfaces of condensate. Microscopy was performed as described above. The presented data were collected over >3 days with multiple condensation reactions per day.

### Analysis

Image analysis was performed semi-automatically using custom scripts. First, an ROI was manually created to include only the

inside of condensates, but excluding the area at least 1 μm inwards from the contour of condensates in order to remove microspheres close to the condensate-water interface. Areas with microsphere aggregates were also excluded from ROIs. Automatic single particle tracking was performed using a custom script and the TrackMate plugin in Fiji (Schindelin et al, 2012; Tinevez et al, 2017). Following the tracking, each movie was manually checked for inappropriate data as follows. Trajectories involving microsphere aggregates were further removed. Any movies with apparent linear movement due to condensate fusion or wetting events were either entirely discarded, or only the affected frames were disregarded for analysis. Using custom scripts, we calculated ensemble time-averaged mean-squared displacement (MSD), meaning all trajectories from each condensate were pooled together without time dependence. Diffusion coefficient (D) and exponent α were calculated by fitting MSD to the 2D diffusion model MSD = $4Dt^\alpha$. Using the Stokes–Einstein–Sutherland equation $D = k_BT/6\pi\eta r$, we estimated the viscosity η. The reported values are mean ± 95% CI.

## Fluorescence recovery after photobleaching (FRAP) assay

The condensation of full-length PGL-3 or D1-D2 was induced at 5 μM PGL-3, 125 mM NaCl. The condensates were allowed to settle on the coverslip surface. Time-lapse images were captured every 1 s. After five frames, an area of 1 μm in diameter within the condensate was photobleached. Custom scripts were used for analysis. The ratio of unbleached and bleached area was calculated for each timepoint. For each trajectory, the timeseries was rescaled so that the means of the prebleach ratio equals 1 and the ratio right after photobleaching equals 0.

## Crosslinking mass spectrometry (XL-MS)

Crosslinking reaction of PGL-3(FL) was performed by combining 7.5 μM protein in 25 mM HEPES, pH 7.5, 125, or 500 mM NaCl with disuccinimidyl dibutyric urea (DSBU, Thermo Fisher) crosslinker to a 2 mM final concentration. DSBU crosslinks K, S, T, and Y residues to each other. After quenching, proteins were digested, and the resulting peptides were desalted and reconstituted in 0.1% formic acid for separation and analysis on a UltiMate3000 UHPLC system (Thermo Fisher) coupled with a Q-Exactive HF-X Orbitrap mass spectrometer (Thermo Fisher). Spectra were searched in Scout (Clasen et al, 2024). Detailed experimental and analysis protocols are provided below.

### Sample preparation
Full-length PGL-3 was thawed and centrifuged at 21,000×g for 5 min to remove large particles. The protein solution was diluted to 7.5 μM PGL-3, 125 mM (permissive for phase separation), or 500 mM NaCl (non-permissive for phase separation), 25 mM HEPES, pH 7.5 at a 100 μL scale. The diluted protein solution was incubated at room temperature for 5 min before adding 2 μL of disuccinimidyl dibutyric urea (DSBU, Thermo Fisher) from a 100 mM stock in DMSO (f.c. 2 mM). The reaction was incubated at room temperature on an end-over-end rotator for 1 h (10 rpm). Crosslinking reaction was quenched by the addition of Tris-HCl, pH 7.5 (f.c. 20 mM), and incubated at room temperature for 30 min.

Crosslinked samples were denatured by the addition of solid urea (96.1 mg, f.c. 8 M, Sigma-Aldrich) and ammonium bicarbonate, pH 8.0 (Ambic, f.c. 100 mM, Acros Organics), and diluted to a final volume of 200 μl with Optima LC/MS grade water. The samples were reduced by the addition of dithiothreitol (DTT, f.c. 5 mM, Sigma-Aldrich) and incubation for 30 min at 30 °C with agitation (700 rpm) on a benchtop thermomixer (Eppendorf). Samples were alkylated by the addition of iodoacetamide (IAA, f.c. 15 mM, Acros Organics) for 45 min at room temperature without agitation in the dark. DTT (f.c. 5 mM) was then added to quench excess IAA, and the samples were incubated for 5 min at room temperature. The samples were diluted by the addition of 3 volumes of 100 mM Ambic, pH 8.0, to reduce the final concentration of urea to 2 M. The samples were digested by trypsin (1:50 enzyme:protein w/w ratio, Pierce) and incubated at 25 °C, 700 rpm, overnight (~16 h) on a thermomixer. The next day, the digested samples were acidified with Optima LC/MS grade 1% trifluoroacetic acid (TFA, Fisher Chemical) and desalted using C18 Sep-Pak cartridges (Waters). The acidified digests were diluted to 1 mL final volume using Buffer A [0.5% TFA in Optima LC/MS water]. C18 cartridges were placed on a vacuum manifold, conditioned twice using 1 ml Buffer B [0.5% TFA, 80% Optima LC/MS grade acetonitrile], followed by equilibration (four times with 1 mL) with Buffer A. Acidified digests were loaded onto the column under a reduced vacuum (1 mL/min), and then washed with Buffer A (four times with 1 mL). The cartridges were placed on 15-mL Falcon tubes, and samples were eluted by centrifugation at 350 rpm for 5 min in a 5910R centrifuge (Eppendorf). The samples were then transferred to a fresh tube and then reduced to dryness using a Vacufuge centrifugal concentrator (Eppendorf). Dried samples were stored at −80 °C until further analysis.

### LC-MS/MS
The LC-MS/MS experiments utilized an UltiMate3000 UHPLC system (Thermo Fisher) coupled with a Q-Exactive HF-X Orbitrap mass spectrometer (Thermo Fisher). To prepare samples, the dried peptides were vigorously resuspended in a 0.1% formic acid in Optima LC/MS grade water by vortexing and sonication. The peptide concentration was measured with Nanodrop One$^C$ microvolume UV–vis spectrophotometer (Thermo Fisher Scientific), and typically ~1 μg of the peptide was injected onto the LC-MS/MS system for each sample in triplicate. The remaining LC-MS/MS methods and parameters follow as described previously (Faustino et al, 2023).

### Crosslink search
Scout (Clasen et al, 2024) v. 1.5.1 was used to search for the crosslinked peptides using a database comprising just the PGL-3 full-length sequence. DSBU_KSTY crosslinker was selected in Scout with default settings. The peptide length was set to a minimum of 6 and a maximum of 60 residues. The minimum peptide mass was set to 500 Da, and the maximum peptide mass was set to 6000 Da. The precursor mass tolerance was set to 10.0 ppm, and the fragment ion precision was set to 20.0 ppm. Digestion enzyme was set to be fully specific trypsin, with cleavage sites after Lys and Arg and cleavage blocked by Pro, and a maximum of three missed cleavages were allowed. DSBU crosslinker mass of 196.0848 Da and MS-cleavable fragments of amine (light, 85.0527 Da) and isocyanate (heavy, 111.0320 Da) with mass shift of 25.9792 Da were set, with residue specificity towards KSTY residues. Static modifications included the carbamidomethylation of cysteine (Delta mass 57.0214 Da), and oxidation of methionine was used as a variable modification (Delta

mass 15.9949 Da). The 2% FDR cut-off was set to filter the crosslink spectral matches (CSMs) at the peptide and residue pair levels. All identifications are summarized in Dataset EV1.

### Classification of crosslink compatibility to published monomer/dimer structures

Because the previous works reported D1 and D2 dimer structures of *C. elegans* PGL-1, PGL-3 SWISS-MODEL models were used to map detected crosslinks to structures (Waterhouse et al, 2018). The SWISS-MODEL models were templated on 5w4a.1.A (Aoki et al, 2021), for D1 dimer and 5cv1.1 (Aoki et al, 2016) for D2 dimer. PGL-1 and PGL-3 are highly conserved with 68.4% identity and 84.0% similarity for D1, and 69.5% identity and 84.4% similarity for D2 (Appendix Fig. S1A,B). The RMSD of published PGL-1 structures and PGL-3 SWISS-MODEL structures were very low (0.066 for D1 and 0.069 for D2 (Fig. EV3A)). In PyMOL using custom scripts, Cα-Cα distances for all crosslinks were measured on the monomers and dimers to obtain intra- and inter-chain distances, respectively. Then, crosslinks below or equal to 30 Å in distance were classified 'compatible' with the tested structure and the rest "incompatible". We used 30 Å as a cut-off based on the length of DSBU. This analysis yields five categories for crosslinks: "compatible only with intra-chain" (i.e., both crosslink sites on the same chain), "compatible only with inter-chain" (i.e., crosslink sites on different chains), "compatible with both intra and inter chains" (i.e., crosslink sites on either the same or different chains), "not compatible with either", and "no structural information". Alpha-Fold3 was used to generate D1-D2 oligomers. The analysis of compatibility was done similarly to above.

### Crosslink abundance and frequency analysis

In Fig. 4A, all unique residue pairs are shown. In Fig. 4B, the total number of crosslinked peptide-spectrum matches (XSMs) associated with crosslinks between the indicated domains are shown, summing across the three replicates. In Fig. 4C, the number of XSMs associated with crosslinks between the indicated domains was divided by the total number of XSMs to generate frequencies. In Fig. EV3B, the unique residue pairs are shown that were identified in both conditions or only in one.

### FA crosslinking assay

D1 and D2 were trace-labelled with DyLight 488 and imaged in the Cy2 channel. IDR, RGG and KGG were trace-labelled with Alexa Fluor 647 and imaged in the Cy5 channel. PGL-3 fragments were diluted to 2 μM in 125 mM NaCl, 25 mM HEPES, pH 7.5. For protein mixtures, each protein was diluted to 2 μM. Diluted protein solutions were incubated at room temperature for 10 min. Formaldehyde(FA) (Fisher Scientific) was diluted to 2% in 125 mM NaCl, 25 mM HEPES, pH 7.5. Equal volumes of 2% FA were added to 2 μM PGL-3 solutions and mixed by pipetting up and down to reach 1 μM PGL-3 domains and 1% FA. Following the incubation at room temperature for 20 min, the reaction was quenched by f.c. 2.5 mM Tris, 20 mM glycine for 5 min. Home-made SDS sample buffer and final 5% BME (Sigma) was added and boiled for 7 min at 70 °C and SDS-PAGE was performed. We used a home-made SDS sample buffer without dyes to increase sensitivity. The composition of the SDS sample buffer at 1x was as follows: 106 mM Tris-HCl, 141 mM Tris base, 2% SDS, 0.51 mM EDTA,

and 2.5% glycerol. After the electrophoresis, gels were quickly rinsed in distilled water and imaged on Amersham Typhoon (Cytiva) for Cy2 and Cy5 channels.

### Pelleting assay

#### Tube preparation
To minimize protein absorption to polypropylene tubes, 5 mg/ml BSA were placed in tubes and incubated at room temperature overnight. Before use, the tube was rinsed three times with reaction buffer [125 mM NaCl, 25 mM HEPES, pH 7.5] to remove residual BSA.

#### Experiment
In-trans component (unlabeled IDR, RGG, and KGG) and Alexa Fluor 647 trace-labeled PGL-3(D1-D2) were diluted to 6 μM each in 125 mM NaCl, 25 mM HEPES, pH 7.5 and incubated at ambient temperature 18–19 °C for 20 min. The protein mixture was centrifuged at 21,300×g for 20 min at 19 °C using a temperature-controlled centrifuge. The supernatant was collected by pipetting, and the pellet was resuspended in 1xLDS sample buffer (Thermo Fisher). The pellet and supernatant fractions were run on SDS-PAGE. Gels were quickly rinsed in distilled water and imaged on Amersham Typhoon (Cytiva) for the Cy5 channel. Analysis was performed on Fiji (Schindelin et al, 2012), and the PGL-3 D1-D2 band intensities were measured using a published protocol (Ohgane and Yoshioka, 2019), and the fraction of D1-D2 in the pellet was determined as pellet band intensity divided by the sum of pellet band and supernatant band intensities.

### Biolayer interferometry (BLI)

BLI experiments were performed using the Octet Red96 BLI instrument (Sartorius). PGL-3 fragments were biotinylated using EZ-Link™ NHS-PEG12-Biotin (Thermo Fisher) following the manufacturer's protocol. For measuring the interaction between RNA and the RGG region, a 30-nt-long poly(rU) biotinylated at the 5' end was synthesized(IDT). Biotinylated molecules were immobilized on streptavidin-coated sensor tips (Sartorius). In the association steps, the loaded sensor tips were exposed to indicated concentrations of non-biotinylated PGL-3, and the dissociation was carried out in buffer. The buffer compositions for all BLI experiments were 125 mM NaCl, 0.01 mg/ml BSA, 0.02% Tween-20, 25 mM HEPES, pH 7.5. All analyses were done using custom scripts. assuming a 1:1 binding model.

### Partition coefficient determination

#### Experiment
We used D1-D2 labelled with DyLight 488 and RGG, IDR, KGG, scrambled RGG labelled with Alexa Fluor 647, and *nos-2* labelled with Alexa Fluor 546. For free dye, Alexa Fluor 647 NHS ester was quenched with glycine. Coverslips were coated with PEG-silane as described here (Alberti et al, 2018) to prevent wetting and aid image analysis. For Fig. 5B,C, 5 μM of each protein was mixed. For Fig. 6A,B, 4 μM of each protein at the indicated RNA concentration was used. For calculating in vivo stoichiometric concentration of RNA, we used values reported here (Saha et al, 2016): 0.68 μM PGL-3 and 50 ng/μl mRNA, assuming average length of 1.5 kb. This converts to 294.1 ng/μl RNA for 4 μM PGL-3. The *nos-2* mRNA is 1.2 kb in length.

## Analysis

Custom scripts were used for analysis. We trained a classifier for semantic segmentation using the sklearn library. The classifier was trained using manually labelled masks, into three classes: "condensates", "out-of-focus condensates", and "background", in the D1-D2 channel using select images from the analysed dataset. After running segmentation on the D1-D2 channel, the "condensate" class was used to define the dense phase of each condensate. For each condensate, the periphery of the 'condensate' class was defined as 'dilute phase' to minimize the effect of uneven illumination. Finally, partition coefficients were calculated as a ratio between the median of two regions, the median of the dense phase/the median of the dilute phase in the fluorescence channel of interest (Appendix Fig. S5E)

## Mass photometry

Mass photometry was performed on TwoMP (Refeyn) using AcquireMP and DiscoverMP (Refeyn) software. Unlabeled PGL-3(D1-D2) was centrifuged at 21,000×*g* for 5 min to remove large particles and then diluted to fourfold higher concentrations of final protein concentrations in 125 mM NaCl, 25 mM HEPES, pH 7.5. The diluted protein was incubated at room temperature for at least 5 min. The mass photometry experiment was performed as described by the manufacturer's protocol. Briefly, 12 µl of 125 mM NaCl, 25 mM HEPES, pH 7.5, was placed on the imaging well to focus and 4 µl of diluted proteins were mixed in by pipetting up and down. The video recording was started briefly after the final dilution. Mass calibration was performed similarly, using sweet potato β-amylase (sigma).

## Blue-native PAGE

PGL-3(D1) and PGL-3(D2) were diluted to 4 µM PGL-3, 125 mM NaCl, and incubated on ice for 10 min. Proteins were then diluted to 0.5, 1, or 2 µM using buffer containing NativePAGE™ Sample Buffer (Thermo Fisher) and immediately loaded onto Native-PAGE™ 4–16% Bis-Tris (Thermo Fisher). Electrophoresis was performed following the manufacturer's protocol. The protein gels were extensively destained and then silver-stained using Pierce™ Silver Stain Kit (Thermo Fisher).

## Data availability

The mass spectrometry proteomics data have been deposited to the ProteomeXchange Consortium via the PRIDE (Perez-Riverol et al, 2025) partner repository with the dataset identifier PXD064592. The source data of this paper are collected in the following BioImage Archive record: S-BIAD2412.

The source data of this paper are collected in the following database record: biostudies:S-SCDT-10_1038-S44319-026-00730-7.

## Peer review information

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

## Acknowledgements

We thank Dr. Philip Mortimer at the JHU Mass Spectrometry Core facility and Dr. Mario A. Bianchet at the JHU Macromolecular Biophysics Core for technical assistance. We also thank the Spangler lab at JHU for Octet maintenance. We thank the Baltimore Worm Club and the Seydoux lab for helpful insights and discussions. RK is funded by the XDBio program at Johns Hopkins University and the Funai Foundation for Information Technology. PS acknowledges support from the Albstein Foundation for brain research. SDF acknowledges support from the NIH Director's New Innovator Award (DP2-GM140926), from the National Science Foundation (MCB-2045844), from a Camille Dreyfus Teacher-Scholar Award, and from a Sloan Fellowship. GS is an investigator of the Howard Hughes Medical Institute.

## Author contributions

**Rimpei Kuroiwa**: Conceptualization; Resources; Data curation; Software; Formal analysis; Validation; Investigation; Visualization; Methodology; Writing—original draft; Writing—review and editing. **Piyoosh Sharma**: Data curation; Investigation; Methodology; Writing—review and editing. **Andrea A Putnam**: Conceptualization; Resources; Methodology; Writing—review and editing. **Stephen D Fried**: Supervision; Funding acquisition; Methodology; Project administration; Writing—review and editing. **Geraldine Seydoux**: Conceptualization; Supervision; Funding acquisition; Methodology; Writing—original draft; Project administration; Writing—review and editing.

Source data underlying figure panels in this paper may have individual authorship assigned. Where available, figure panel/source data authorship is listed in the following database record: biostudies:S-SCDT-10_1038-S44319-026-00730-7.

## Disclosure and competing interests statement

The authors declare no competing interests.

# Expanded View Figures

**Figure EV1.  The IDR-RGG region does not undergo phase separation.**

(**A**) Protein gel stained with Coomassie Brilliant Blue G250 showing the recombinant PGL-3 derivatives used in the condensation assays shown in Fig. 1C, D. About 1 µg protein per well was loaded. (**B**) Table listing the region and the molecular weight for each PGL-3 constructs used in this study. (**C**) DIC micrographs of unlabeled PGL-3 and its derivatives of indicated protein concentrations at 67.5 or 125 mM NaCl. Background pattern visible in all panels (and also in DIC panels in (**D**, **E**)) originates from microscope internal parts. Scale bar = 20 µm. (**D**) DIC micrographs of unlabeled PGL-3 IDR-RGG at 101.6 µM PGL-3(IDR-RGG), 125 mM NaCl and varying concentrations of *nos-2* mRNA. Scale bar = 20 µm. (**E**) DIC micrographs of unlabeled wild-type and D1 dimerization mutant of full-length PGL-3 tested for condensation at the indicated protein and salt concentrations. Note the robust condensation of wild-type PGL-3, which is not observed for the mutant. PGL-3(R123E, K126E, K129E) is full-length PGL-3 with mutations in the D1 domain, previously shown to interfere with dimerization in PGL-1 (Aoki et al, 2021). Scale bar = 20 µm.

▶

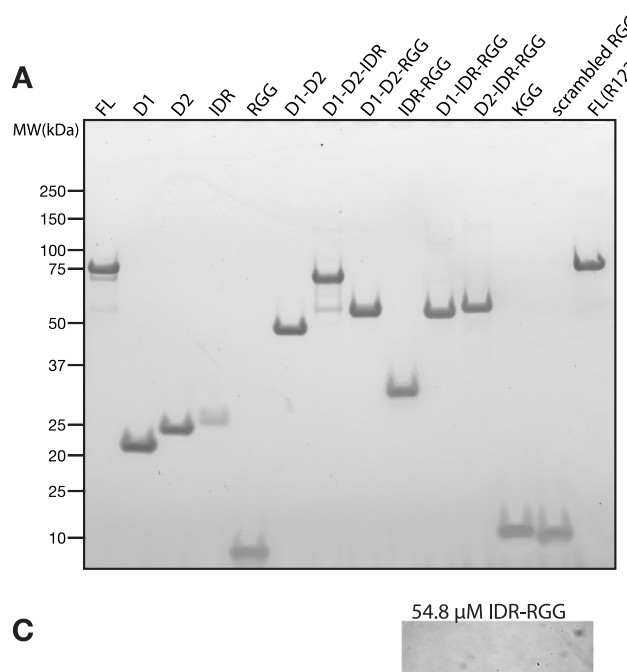

**B**

| Construct | Residues | MW (kDa) |
|---|---|---|
| FL | 1-693 | 74.8 |
| D1 | 1-212 | 23.5 |
| D2 | 205-452 | 28.0 |
| IDR | 448-622 | 18.1 |
| RGG | 623-693 | 6.76 |
| D1-D2 | 1-447 | 50.0 |
| D1-D2-IDR | 1-622 | 68.1 |
| D1-D2-RGG | 1-447::623-693 | 56.78 |
| IDR-RGG | 448-693 | 24.8 |
| D1-IDR-RGG | 1-212::448-693 | 48.3 |
| D2-IDR-RGG | 205-693 | 51.9 |
| KGG | 623-693(all R to K) | 6.39 |
| scrambled RGG | 623-693(scrambled) | 6.76 |

**C**

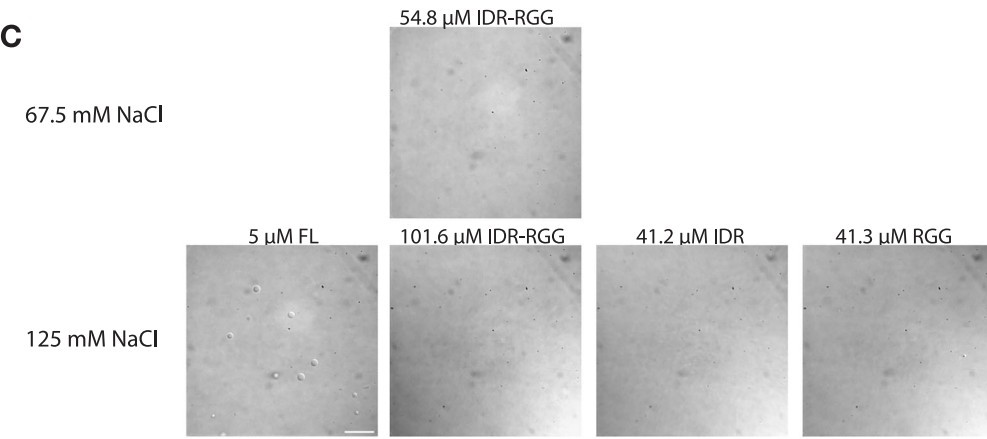

**D**

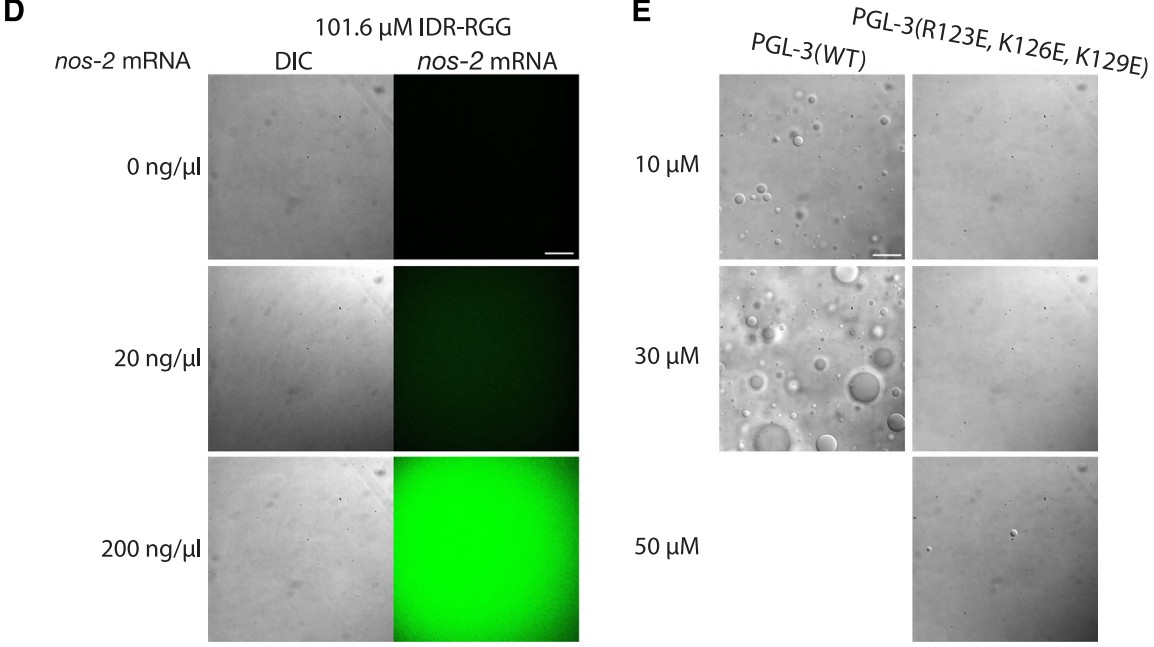

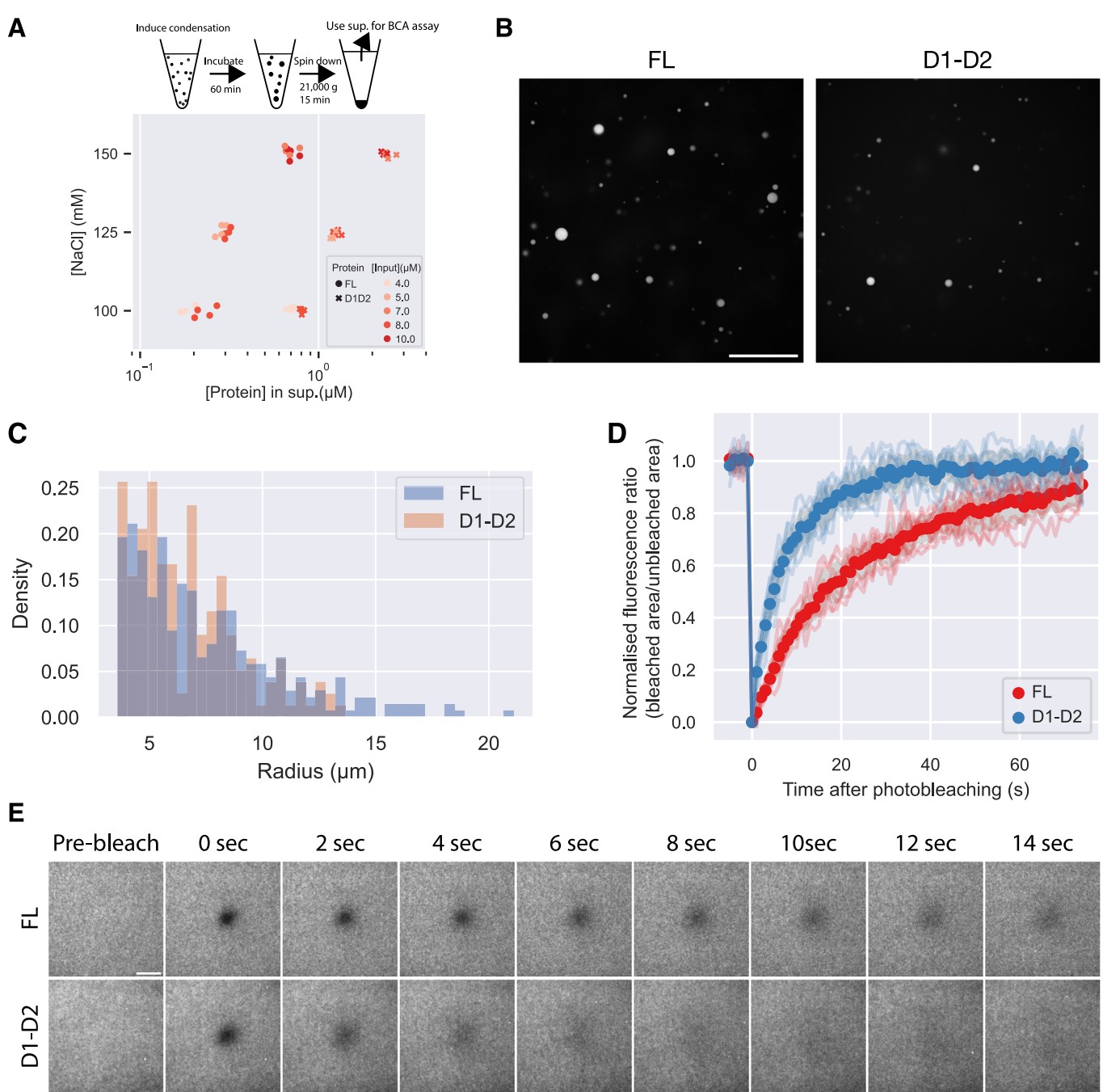

**Figure EV2. Further comparison of PGL-3 FL and D1-D2 condensates.**

(A) Top: Schematic describing the procedure of $c_{sat}$ estimation by $c_{dil}$ measurement. Bottom: Plot showing $c_{dil}$ of full-length PGL-3 and D1-D2 at varying protein/salt concentrations permissive for condensation. (B) Representative photomicrograph showing condensates of full-length PGL-3 or D1-D2 at 5 μM protein, 125 mM NaCl. Scale bar = 50 μm. (C) Density histogram showing size distribution of PGL-3 condensates. $N = 314$ for full-length PGL-3, $N = 178$ for D1-D2. (D) Plot showing FRAP of full-length and D1-D2 condensates. Thin lines show individual traces from single condensates, dots show means, and the gray areas show standard deviation. $N > 9$. (E) Representative time-lapse photomicrographs showing fluorescence recovery after photobleaching (FRAP) of a 1 μm diameter circular area within a condensate. Note the faster recovery in the D1-D2 condensate. Scale bar = 5 μm.

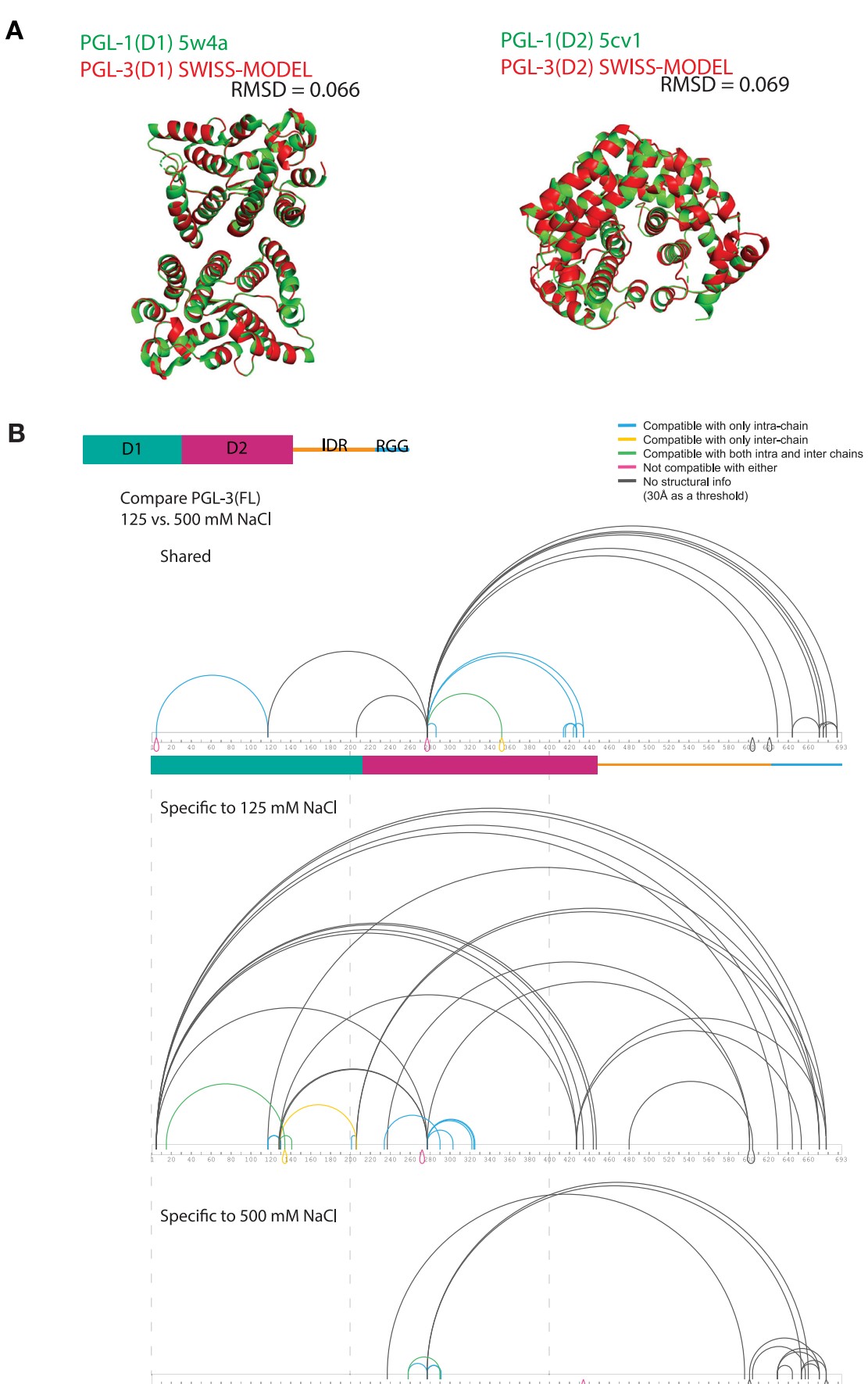

**Figure EV3. Overview of condition-specific crosslinks.**

(A) Alignment of D1 (residues 1–212) and D2 (residues 205–447) between published crystal D1 and D2 dimer structures of PGL-1 (Aoki et al, 2016, 2021) and SWISS-MODEL(Waterhouse et al, 2018) structures of PGL-3. (B) Connectogram comparing crosslinking profiles of PGL-3 at 125 and 500 mM NaCl. Top: Shared between two conditions. Middle: Specific to 125 mM NaCl. Bottom: Specific to 500 mM NaCl.

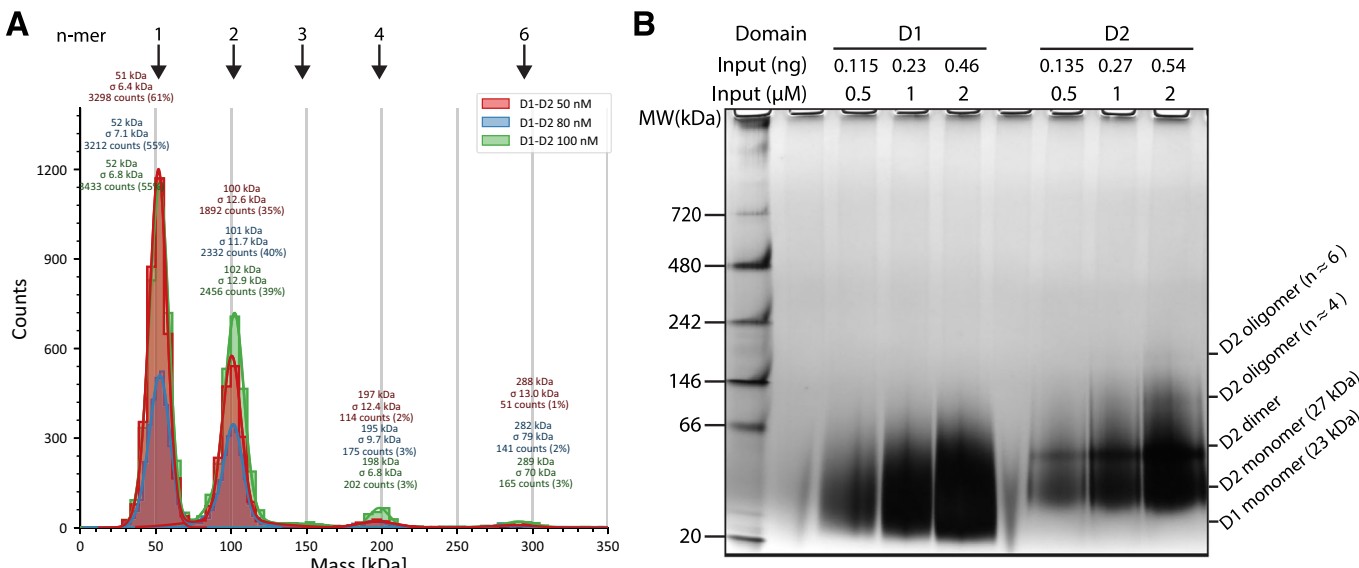

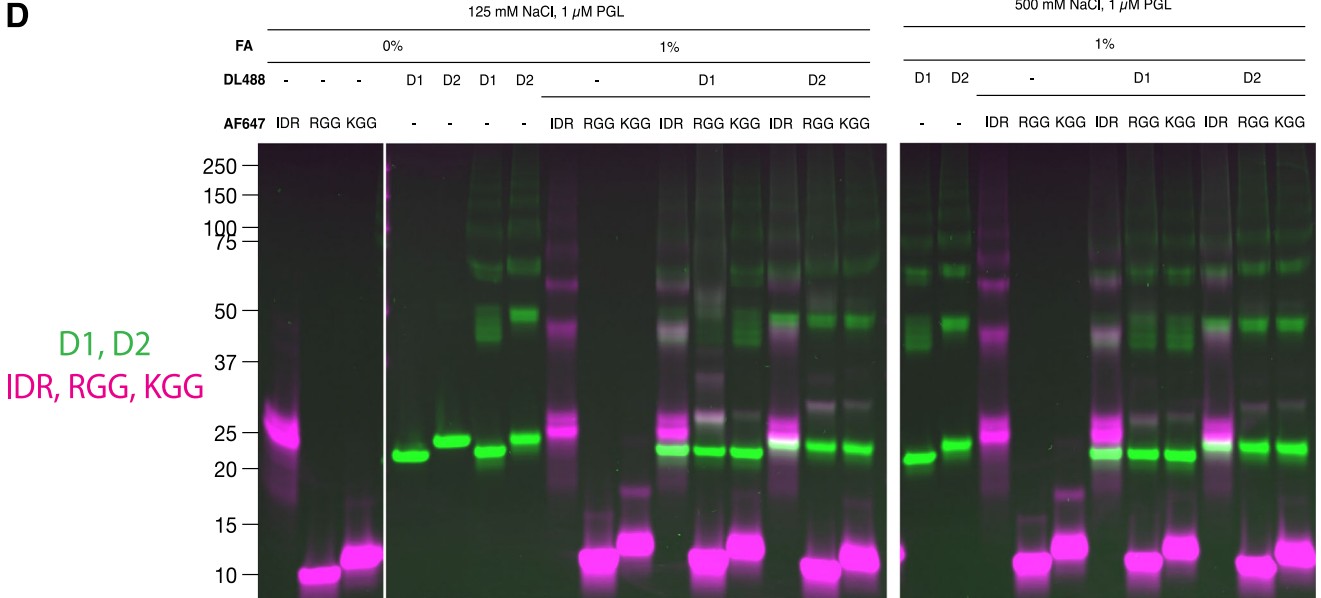

**Figure EV4.  PGL-3 D1 and D2 crosslinks to the RGG region.**

(A) Mass histogram of D1-D2 oligomers in 125 mM NaCl solution, measured by mass photometry. D1-D2 was diluted by fourfold to reach the indicated concentrations in imaging wells and the videos were immediately recorded on a TwoMP (Refeyn). (B) Blue-native PAGE, visualized by silver stain, of single-domain constructs, running D1 and D2, of PGL-3 at indicated concentrations. (C) Sequence alignment among the native RGG peptide, the KGG peptide and the scrambled RGG peptide. RGG triplets and RG doublets are highlighted in magenta and yellow, respectively. (D) Dual-color fluorescent image of the gel shown in Fig. 5A. Two channels overlaid in color (DyLight 488 in green and Alexa Fluor 647 in magenta).

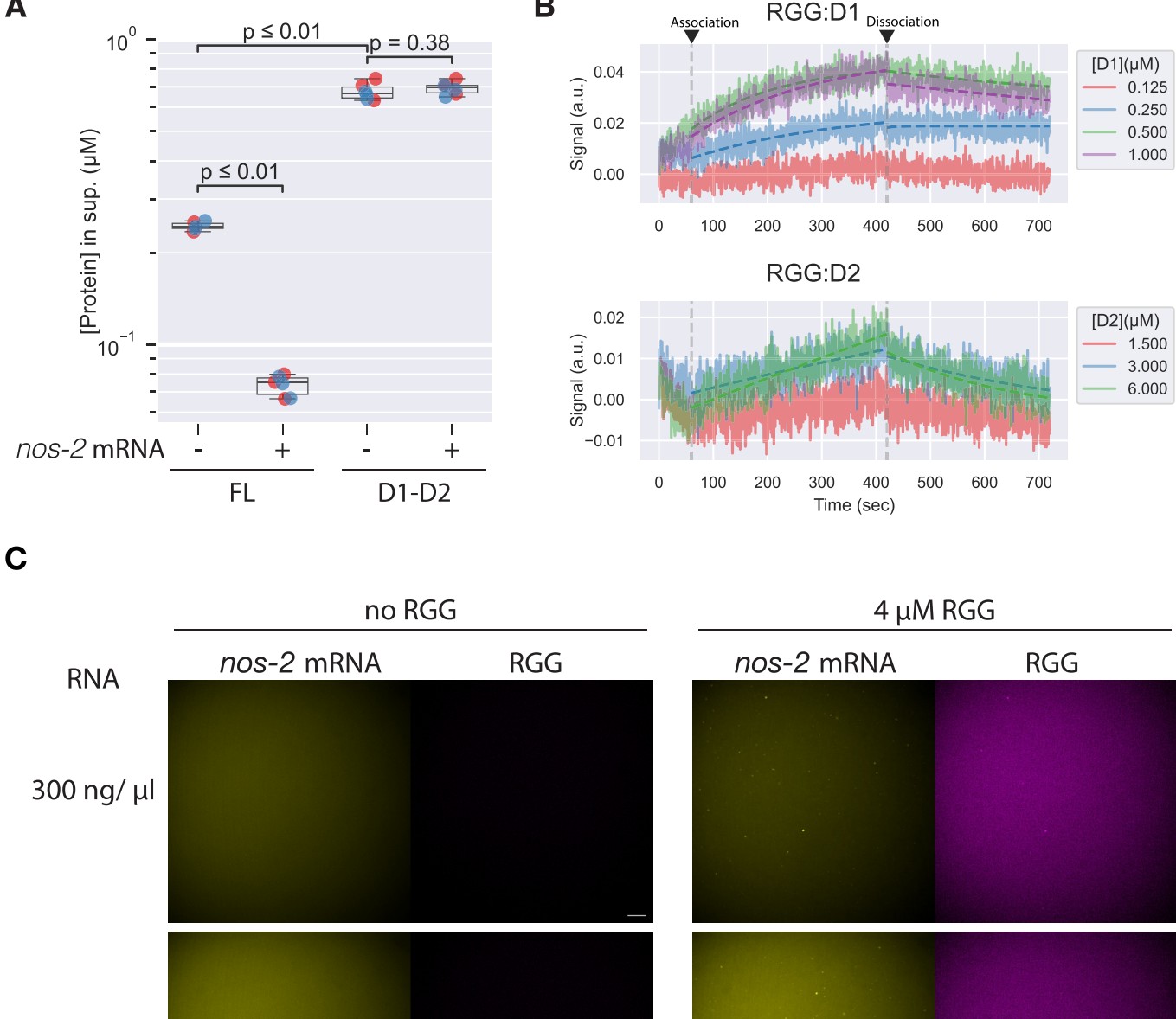

**Figure EV5.   Contributions of RGG domain to condensation.**

(A) Plot showing dilute phase protein concentrations of condensation reactions carried out with 1.5 μM PGL-3, 125 mM NaCl in the presence or absence of 110 ng/μl *nos-2* mRNA. Different colors represent data from different days. In the overlaying box plots, lines inside boxes show median (Q2), box bounds are quartiles (Q1 and Q3), and whiskers show the range of data excluding outliers identified by the Turkey method. Technical replicates. $N = 6$. *P* values calculated by Wilcoxon rank-sum test with Benjamini–Hochberg adjustment for multiple comparisons. *P* values = 0.002165 (FL vs FL, RNA), 0.002165 (FL vs. D1-D2), 0.3768 (D1-D2 vs. D1-D2, RNA). (B) BLI sensorgrams for testing individual domains, D1 and D2, bind to the RGG peptide. At the indicated timepoints, the RGG peptide immobilized on the sensor tip was allowed to associate with indicated concentrations of D1 or D2. Dissociation steps were carried out in a buffer without D1 or D2. Note the differences in input concentrations between D1 and D2. (C) Fluorescence micrographs showing distribution of *nos-2* mRNA (yellow) and the RGG peptide (magenta) at indicated concentrations, without D1-D2. Note RNA and RGG form co-aggregates only when the two components are mixed. Scale bar = 10 μm.

