## [Peer Review File · EMBO Reports]

Phase separation of PGL-3 driven by structured domains that oligomerize and interact with RGG motifs

Rimpei Kuroiwa, Piyoosh Sharma, Andrea Putnam, Stephen Fried, and Geraldine Seydoux

Corresponding author(s): Geraldine Seydoux (gseydoux@jhmi.edu)

Review Timeline:

Submission Date:	10th Jul 25
Editorial Decision:	4th Sep 25
Revision Received:	21st Nov 25
Editorial Decision:	8th Jan 26
Revision Received:	14th Jan 26
Accepted:	12th Feb 26

Transaction Report:

Dear Geraldine,

Thank you for the submission of your research manuscript to our journal. Three referees agreed to review your manuscript. So far, we have received two referee reports that are copied below. Given that both referees are in fair agreement that you should be given a chance to revise the manuscript, I have decided to proceed with these two reports.

Given the constructive and very positive referee comments, I would thus like to invite you to revise your manuscript with the understanding that the referee concerns (as detailed above and in their reports) must be fully addressed and their suggestions taken on board. Please address all referee concerns in a complete point-by-point response. Acceptance of the manuscript will depend on a positive outcome of a second round of review. It is EMBO Reports policy to allow a single round of revision only and acceptance or rejection of the manuscript will therefore depend on the completeness of your responses included in the next, final version of the manuscript.

We realize that it is difficult to revise to a specific deadline. In the interest of protecting the conceptual advance provided by the work, we recommend a revision within 3 months (December 3rd). Please discuss the revision progress ahead of this time with the editor if you require more time to complete the revisions.

I am also happy to discuss the revision further via e-mail or a video call, if you wish.

=====

IMPORTANT NOTE:

We perform an initial quality control of all revised manuscripts before re-review. Your manuscript will FAIL this control and the handling will be delayed IN CASE the following APPLIES:

- 1) A data availability section providing access to data deposited in public databases is missing. If you have not deposited any data, please add a sentence to the data availability section that explains that.
- 2) Your manuscript contains statistics and error bars based on $n=2$. Please use scatter blots in these cases. No statistics should be calculated if $n=2$.

=====

- 1) a .docx formatted version of the manuscript text (including legends for main figures, EV figures and tables). Please make sure that the changes are highlighted to be clearly visible.
- 2) individual production quality figure files as .eps, .tif, .jpg (one file per figure). Please download our Figure Preparation Guidelines (figure preparation pdf) from our Author Guidelines pages <https://www.embopress.org/page/journal/14693178/authorguide> for more info on how to prepare your figures.
- 3) a .docx formatted letter INCLUDING the reviewers' reports and your detailed point-by-point responses to their comments. As part of the EMBO Press transparent editorial process, the point-by-point response is part of the Review Process File (RPF), which will be published alongside your paper.
- 4) a complete author checklist, which you can download from our author guidelines (<<https://www.embopress.org/page/journal/14693178/authorguide>>). Please insert information in the checklist that is also reflected in the manuscript. The completed author checklist will also be part of the RPF.

5) Please note that all corresponding authors are required to supply an ORCID ID for their name upon submission of a revised manuscript (<<https://orcid.org/>>). Please find instructions on how to link your ORCID ID to your account in our manuscript tracking system in our Author guidelines

(<<https://www.embopress.org/page/journal/14693178/authorguide#authorshipguidelines>>)

6) We replaced Supplementary Information with Expanded View (EV) Figures and Tables that are collapsible/expandable online. A maximum of 5 EV Figures can be typeset. EV Figures should be cited as 'Figure EV1, Figure EV2' etc... in the text and their respective legends should be included in the main text after the legends of regular figures.

7) Suggested wording for the "Data Availability" section: "The [structural coordinates | microarray | mass spectrometry] data from this publication have been deposited to the [name of the database] database [URL] and assigned the identifier [accession | permalink | hashtag].". Please note that we need links that resolve directly to the dataset. Please only refer to data deposited in external repositories and remove the information on data available in the paper ('All other data needed to evaluate the conclusions in this paper are present in the paper and/or the SI Appendix.")

Additional information on source data and instruction on how to label the files are available

<<https://www.embopress.org/page/journal/14693178/authorguide#sourcedata>>

10) Figure legends and data quantification:

- the name of the statistical test used to generate error bars and P values,
 - the EXACT p-values,
 - the number (n) of independent experiments (please specify technical or biological replicates) underlying each data point,
 - the nature of the bars and error bars (s.d., s.e.m.)
- If the data are obtained from n {less than or equal to} 5, show the individual data points in addition to the SD or SEM.
- If the data are obtained from n {less than or equal to} 2, use scatter blots showing the individual data points.

11) Our journal encourages inclusion of *data citations in the reference list* to directly cite datasets that were re-used and obtained from public databases. Data citations in the article text are distinct from normal bibliographical citations and should directly link to the database records from which the data can be accessed. In the main text, data citations are formatted as follows: "Data ref: Smith et al, 2001" or "Data ref: NCBI Sequence Read Archive PRJNA342805, 2017". In the Reference list, data citations must be labeled with "[DATASET]". A data reference must provide the database name, accession number/identifiers and a resolvable link to the landing page from which the data can be accessed at the end of the reference. Further instructions are available at <<https://www.embopress.org/page/journal/14693178/authorguide#referencesformat>>.

12) All Materials and Methods need to be described in the main text using our 'Structured Methods' format. According to this format, the Methods section includes a Reagents and Tools Table (listing key reagents, experimental models, software and relevant equipment and including their sources and relevant identifiers) followed by a Methods and Protocols section describing

the methods, ideally using a step-by-step protocol format. The aim is to facilitate adoption of the methodologies across labs. Please download and fill our Reagents and Tools Table template (.docx), which you can find in our author guidelines: <https://www.embopress.org/page/journal/14693178/authorguide#structuredmethods>.

An example of a Method paper with Structured Methods can be found here: <https://www.embopress.org/doi/10.15252/msb.20178071>.

13) As part of the EMBO publication's Transparent Editorial Process, EMBO Reports publishes online a Review Process File to accompany accepted manuscripts. This File will be published in conjunction with your paper and will include the referee reports, your point-by-point response and all pertinent correspondence relating to the manuscript.

Kind regards,

Martina

=====

Referee #1:

The paper by Kuroiwa et al. re-examines the molecular basis of phase separation of PGL-3, a component of the P granules in *C. elegans*. This is an exciting manuscript, with clear logic and experimentally well done. The authors use a comprehensive approach-including in vitro reconstitution, mutagenesis, crosslinking mass spectrometry, and biophysical assays-to dissect the respective contributions of the structured D1-D2 domains, RGG motifs, and the IDR to PGL-3 condensation. Briefly, Kuroiwa et al. systematically test the roles of different PGL-3 domains and their combinations to assess the propensity for phase separation, showing that the D1-D2 domains drive this process. The authors then identify distinct properties between the condensates formed by full-length and D1-D2 constructs, as well as the enhancement provided by RGG domains. Based on these findings, the authors identify potential interacting regions between these domains and the propensity for oligomerization. Finally, they show that RNA competes with D1-D2 for the RGG repeats in trans. This work uncovers the role of folded domains in forming condensates and advances our understanding of the molecular mechanisms underlying P granule formation. This manuscript is a strong fit for EMBO Reports, and I have only a few minor and major suggestions.

Minor:

- In Figure 1d and e, although the legend mentions the scale bar size, only one image contains a bar, which is almost invisible. I recommend adding scale bars to all images, as it is easier for the reader than repeatedly checking a single panel. Please use clear, visible scale bars. Check for the same issue in all figures (e.g., Fig. 2c, Fig. 3a, Fig. 5b).

- In Fig. 1d, droplets formed by full-length protein in vitro seem smaller than those formed by D1-D2, but this size difference is less obvious in other panels (e.g., Fig. 1e). It would be important to analyze the size distribution of droplets for each PGL-3 fragment tested, especially full-length and D1-D2, where microrheology assays were performed, since condensate size alone can limit bead diffusion.

- In Fig. 5a, please comment on the different band patterns observed between D1 and D2 dimers. D2 has a clear, sharp band as expected for a dimer, but D1 displays several discrete bands. Also, the font size of the labels is too small.

- In Fig. 5a, it appears that the majority of the D1 and D2 protein populations do not oligomerize. This could be due to the extent

of crosslinking or low binding affinity (see comment below).

- It would be beneficial to include a model in the final figure.

Major:

- In Fig. 2c-d, the findings should be further supported with an orthogonal approach. For example, fluorescence recovery after photobleaching (FRAP) assays could be performed.

- In Fig. 2e, the low viscosity of D1-D2 condensates is surprising given that oligomerization of folded domains is the driving mechanism. Measuring the binding affinities between these domains could help provide an explanation (i.e. it would be expected that the respective K_d 's would be rather high - μM range). Given that all is in place for such an experiment, this should be provided.

- In Fig. 6a, the authors observe competition between D1-D2 domains and nos-2 RNA for the RGG repeats in trans, which leads to the formation of aggregates between nos-2 RNA and RGG repeats. However, in Fig. 1e, RNA enhances phase separation of full-length PGL-3 (e.g. when the RGG repeats are in cis), and the authors also found that D1-D2-RGG interactions enhance condensate formation both in cis and trans, it would be informative to weigh these relationships and their contributions to phase separation overall. For instance, how does RNA affect the csat of full length PGL-3? It would be good to get a sense of the relative contributions of RNA-dependent and RNA-independent contributions to phase separation in this system.

- Because overall, RNA binding via the RGG repeats still stimulates phase separation of the full length protein the final sentence of the abstract is a bit misleading: while factually correct (there is an RNA-independent contribution) it ignores the fact that RNA does contribute. Hence a phrase like "...enhanced by RGG repeats both dependent and independent of RNA" would be a more faithful reflection of the results.

Referee #3:

Kuroiwa et al. review August 2025

The authors present a well-motivated study: while the canonical drivers of phase separation (PS) are intrinsically disordered regions (IDRs) and low-complexity regions (often also disordered), emerging evidence indicates that ordered regions can also contribute-and in some cases may be the primary drivers. Using the *C. elegans* P-granule protein PGL-3 as a model, the authors employ an in vitro reconstitution approach to define which regions are necessary and/or sufficient for PS and how these regions interact to promote or inhibit the process. Contrary to prevailing dogma, they find that PGL-3's ordered domains (D1-D2) are not only sufficient but appear to be the principal drivers of PS in this system. The low-complexity RGG region modulates the critical concentration for PS and subtly alters condensate biophysical properties-full-length PGL-3 and D1-D2 condensates differ in viscosity, partly explained by the RGG addition experiments. The IDR linking D1-D2 exerts an inhibitory effect, as D1-D2-IDR constructs do not show PS (but the IDR does appear to multimerize in Figure 5A). Although all data are derived from in vitro systems (as noted in the limitations section), this work compellingly shows that-contrary to predictions from current algorithms-ordered regions can be dominant drivers of PS, suggesting that these algorithms may need revision.

The work is clearly presented and addresses an important conceptual gap. The data are of high quality, though the discussion could be strengthened by deeper mechanistic interpretation and integration of certain results that are currently underexplored.

Major Comments

1. Mechanism of D1-D2 Oligomerization

The manuscript states that "D1 and D2 both have the ability to form oligomers and provide the requisite multivalency for condensation," but the source of this multivalency is not elaborated. Could the data suggest that a valency greater than two is sufficient for PS, with structured domains achieving this via interdomain hydrophobic-core interactions?

Incorporating Structural modeling (e.g., AlphaFold-Multimer), possibly highlighting interacting residues from the crosslinking mass spectrometry data, could strengthen the argument. My exploratory simulations suggest that D1-D2 may be capable of forming hollow shells of variable stoichiometry. I was also intrigued that the condensates in the movie appeared to exhibit a strikingly uniform size distribution, suggesting a potential self-limiting assembly mechanism.

2. Interpretation of Figure 5A

The data in Figure 5A support an interaction between D1-D2 and the RGG region, but the behavior of the IDR in this assay is not addressed. The IDR appears to form high-order multimers that may overlap with D1 or D2 bands, yet in the LLPS assays, the IDR does not self-assemble or co-assemble with D1-D2. This discrepancy warrants discussion.

3. IDR-Mediated Inhibition of PS

The mechanism by which the IDR inhibits D1-D2-driven PS is not explored. An interpretation of potential inhibitory mechanisms would strengthen the discussion.

4. Depth of Discussion

The discussion section largely reiterates results rather than interpreting them. Incorporating more mechanistic insights, speculative models, or connections to the broader PS field would enhance impact. Some experimental findings go undiscussed and should be addressed. Given that RGG motifs can function in trans, could P-granule proteins serve as RGG donors for other constituent proteins that cannot phase separate using structured domains alone? This possibility has interesting physiological implications that merit discussion.

Minor Comments

1. Include scale bars in all microscopy images.
2. Clarify why crowding agents were not used in the in vitro assays.
3. In the abstract (lines 9-10), consider rephrasing "D1-D2 is oligomeric" to "D1-D2 has the potential to self-assemble."
4. The nomenclature for the "IDR" and "RGG" regions is inconsistent and potentially misleading, given that both are disordered. It may be clearer to refer to them as regions rather than domains, reserving the term "domain" for folded, independently stable units.
5. Correct the text at line 116 to refer to Fig. 2C; Figure 2D is not cited.
6. Expand the legends for Figures 3B and 5D to provide sufficient methodological detail.
7. For Figure 5A, clarify whether 1% crosslinking is required to observe multimerization and whether native PAGE could serve as an alternative approach.
8. D1 requires D2 to PS since D1-D2 phase separates in Fig 1D while D1 and D1-IDR-RGG cannot, which suggests that D1:D2 is the minimum necessary interaction for PS with RGG interaction serving a complementary role.

Referee #1:

The paper by Kuroiwa et al. re-examines the molecular basis of phase separation of PGL-3, a component of the P granules in *C. elegans*. This is an exciting manuscript, with clear logic and experimentally well done. The authors use a comprehensive approach-including in vitro reconstitution, mutagenesis, crosslinking mass spectrometry, and biophysical assays-to dissect the respective contributions of the structured D1-D2 domains, RGG motifs, and the IDR to PGL-3 condensation. Briefly, Kuroiwa et al. systematically test the roles of different PGL-3 domains and their combinations to assess the propensity for phase separation, showing that the D1-D2 domains drive this process. The authors then identify distinct properties between the condensates formed by full-length and D1-D2 constructs, as well as the enhancement provided by RGG domains. Based on these findings, the authors identify potential interacting regions between these domains and the propensity for oligomerization. Finally, they show that RNA competes with D1-D2 for the RGG repeats in trans. This work uncovers the role of folded domains in forming condensates and advances our understanding of the molecular mechanisms underlying P granule formation. This manuscript is a strong fit for EMBO Reports, and I have only a few minor and major suggestions.

Response: We thank the reviewer for their positive evaluation and valuable suggestions to improve our manuscript. We have addressed all the suggestions point-by-point.

Minor:

- In Figure 1d and e, although the legend mentions the scale bar size, only one image contains a bar, which is almost invisible. I recommend adding scale bars to all images, as it is easier for the reader than repeatedly checking a single panel. Please use clear, visible scale bars. Check for the same issue in all figures (e.g., Fig. 2c, Fig. 3a, Fig. 5b).

Response: We thank the reviewer for their suggestion. In our revised manuscript, we used longer and thicker scale bars and modified the figure legends accordingly.

- In Fig. 1d, droplets formed by full-length protein in vitro seem smaller than those formed by D1-D2, but this size difference is less obvious in other panels (e.g., Fig. 1e). It would be important to analyze the size distribution of droplets for each PGL-3 fragment tested, especially full-length and D1-D2, where microrheology assays were performed, since condensate size alone can limit bead diffusion.

Response: We apologize for the misleading micrographs. The FL condensates are indeed, on average, larger and more abundant, as expected given its lower c_{sat} . We updated Fig. 1D with more representative images to better reflect this. In addition, as per the suggestion, we quantified the size-distribution of FL and D1-D2 condensates at the same input concentration (Response figure 1A, B; now Fig. EV2B,C, Appendix Fig. S2C in paper). These data demonstrate that FL condensates are larger and more abundant than D1-D2 condensates.

Response figure 1. Full-length PGL-3 form larger condensates on average than D1-D2.

(A) Representative photomicrograph showing condensates of full-length PGL-3 or PGL-3(D1-D2) at 5 μM PGL-3, 125 mM NaCl. Scale bar = 50 μm

(B) Density histogram showing the size distribution of PGL-3 condensate. Total number used for calculating the density was 314 for full-length PGL-3 and 178 for PGL-3(D1-D2).

(C) Count histogram showing the size distribution of PGL-3 condensates. Related to B.

Regarding the microrheology experiments, we excluded trajectories at the condensate periphery (that is, diffused within 1 μm from the condensate contour) to eliminate any possible surface effects. Also, as Movie EV1 shows (entire field of view is within a single droplet), the spatial scales of bead diffusion are smaller than the droplet diameters examined for this experiment, so we expect minimal effects of condensate size on our quantifications. In addition, as suggested, we performed FRAP measurements on FL and D1-D2 condensates and found that FL condensates exhibit slower internal diffusion, consistent with higher viscosity based on the microrheology experiments (Response figure 2; now Fig. EV2D,E in paper).

Response figure 2. Full-length PGL-3 condensates exhibit slower internal diffusion.

(A) Representative time-lapse photomicrographs showing fluorescence recovery after photobleaching (FRAP). For each condensate, 1 μm diameter circular area at the center of field of view was photobleached. Scale bar = 5 μm

(B) Plot showing FRAP of full-length and D1-D2 condensates. Thin lines show individual traces from single condensates; dots show means and the gray areas show standard deviation. $n > 9$.

- In Fig. 5a, please comment on the different band patterns observed between D1 and D2 dimers. D2 has a clear, sharp band as expected for a dimer, but D1 displays several discrete bands. Also, the font size of the labels is too small.

Response: We have enlarged the labels on Fig. 5A. We do not know why D1 and D2 behave differently but could speculate that the different D1 bands may reflect distinct dimer conformations that use different interfaces and hence get crosslinked at different residues. Crosslinked proteins do not always run as expected based on their mass on SDS-PAGE, so dimers that crosslink and “branch” off each other differently could have different electrophoretic mobilities. We chose not to include this discussion in the manuscript

because we have more direct evidence for D1 possessing multiple dimerization interfaces (i.e., XL-MS and mass photometry).

- In Fig. 5a, it appears that the majority of the D1 and D2 protein populations do not oligomerize. This could be due to the extent of crosslinking or low binding affinity (see comment below).

Response: We agree that the low crosslinking efficiency could reflect poor crosslinking conditions and/or low binding affinity. However, we point out that we optimised the FA crosslinking experiment conditions to maintain most of the protein in a monomeric state so as to avoid over-saturation of the oligomer bands and also to minimize protein aggregation.

- It would be beneficial to include a model in the final figure.

Response: We appreciate the suggestion. We added a new figure (Response figure 3; Fig. 7 in paper), that summarises our findings and proposes a model for PGL-3 phase separation.

Response figure 3. Proposed model for PGL-3 phase separation.

Schematics showing the interactions that drive PGL-3 phase separation (PS). The schematic below summarizes the roles of each domain.

Top left panel: D1 and D2 domains are oligomerizing domains and provide the foundation of PGL-3 PS.

Bottom left panel: IDR-RGG have limited self-associative interactions and do not undergo PS.

Middle panel: IDR-RGG region enhances PS by promoting binding interactions between D1-D2 and the RGG region.

Right panel: RNA further enhances PS by binding to the RGG region.

The internal IDR may contain sites for binding to other proteins or for post-translational (PTMs) modifications.

Major:

- In Fig. 2c-d, the findings should be further supported with an orthogonal approach. For example, fluorescence recovery after photobleaching (FRAP) assays could be performed.

Response: We appreciate the suggestion. We performed FRAP experiments and found that the results corroborate the microrheology experiments (Response figure 2 above; Fig. EV2D,E in revised manuscript)

- In Fig. 2e, the low viscosity of D1-D2 condensates is surprising given that oligomerization of folded domains is the driving mechanism. Measuring the binding affinities between these domains could help provide an explanation (i.e. it would be expected that the respective K_d 's would be rather high - μM range). Given that all is in place for such an experiment, this should be provided.

Response: We measured binding kinetics of D1 and D2 domains using bio-layer interferometry (Response figure 4). D1-D2 were immobilized on sensor tips and tested for binding to D1, D2 or D1-D2 in solution. We observed K_D s in the sub μM to μM range and binding off-rates at sub-second timescale.

The observed low viscosity of D1-D2 condensates may seem surprising given the slow off-rates observed by bio-layer interferometry with isolated domains. However, because condensate viscosity is influenced both by binding kinetics and density, it is not straightforward to predict mesoscale material properties from microscopic properties. From the BLI data with D1-D2:D1-D2, we can infer binding kinetics for two binding sites, but additional, possibly weaker, binding sites must exist to achieve PS. We speculate that the binding modes detected by BLI drive oligomerisation, while additional weak interfaces, undetected in those experiments, provide the additional transient valency necessary for PS.

Response figure 4. Kinetics of D1, D2 and D1-D2 binding to D1-D2

BLI sensorgrams showing the interactions, or lack thereof, of D1, D2 and D1-D2 to D1-D2. At the indicated timepoints, the D1-D2 immobilised on sensor tip was allowed to associate to indicated concentrations of D1, D2, or D1-D2. Dissociation steps were carried out in buffer. D1 did not show significant binding to D1-D2. Fitted parameters are shown on the right. Because the association steps of D2 and D1-D2 and the dissociation step of D1-D2 were not well fitted by single mode of binding, we used two independent modes of binding to fit these steps. Opaque solid lines are experimental data and dotted lines are model fits.

- In Fig. 6a, the authors observe competition between D1-D2 domains and nos-2 RNA for the RGG repeats in trans, which leads to the formation of aggregates between nos-2 RNA and RGG repeats. However, in Fig. 1e, RNA enhances phase separation of full-length PGL-3 (e.g. when the RGG repeats are in cis), and the authors also found that D1-D2-RGG interactions enhance condensate formation both in cis and trans, it would be informative to weigh these relationships and their contributions to phase separation overall. For instance, how does RNA affect the c_{sat} of full length PGL-3? It would be good to get a sense of the relative contributions of RNA-dependent and RNA-independent contributions to phase separation in this system.

Response: To explore the relative effect of protein-protein interactions and protein-RNA interactions, we measured the c_{sats} of FL and D1-D2, with and without mRNA (Response figure 5A; now Fig. EV5A). We observed that RNA decreased the c_{sat} of FL but not D1-D2, consistent with the RGG domain being responsible for RNA binding¹. The absolute c_{sat}

difference between D1-D2 and FL was larger than that between FL and FL+RNA. In addition, because we detect robust binding of D1 to RGG, but not D2 to RGG, we compared the binding kinetics of D1 to RGG and RNA to RGG (Response figure 5B, C; now Fig. 6A, Fig. EV5B in the manuscript). We found that k_{on} and k_{off} were both one order of magnitude higher for D1:RGG than for RNA:RGG, consistent with decreased internal diffusivity of full-length PGL-3 condensates when supplemented with mRNA¹. We also found that the affinity for D1 was higher than RNA. However, we emphasize that the biological relevance of these comparisons will require further validations in a physiological context and we have expanded on this topic in the Discussion.

Response figure 5. Quantitative comparison of relative effect of IDR-RGG and RGG to D1-D2 phase separation.

(A) Plot showing the measured protein concentrations of the dilute phase from condensation reactions carried out in $1.5 \mu\text{M}$ PGL-3, 125 mM NaCl with or without $110 \text{ ng}/\mu\text{l}$ nos-2 mRNA. Different colors represent data from two independent days, with 3 replicates per day. Technical replicates. $N = 6$. P-values calculated by Wilcoxon rank-sum test with Benjamini-Hochberg adjustment for multiple comparisons. FL vs FL, RNA: $p = 2.165 \times 10^{-3}$. FL vs. D1-D2: p value = 0.02165 . D1-D2 vs. D1-D2, RNA: p value = 0.3768 .

(B) BLI sensorgrams testing D1 and D2 domain binding to the RGG peptide. At the indicated timepoints, the RGG peptide immobilised on sensor tip was allowed to associate to indicated

concentrations of D1 or D2. Dissociation steps were carried out in buffer without D1 or D2. Note the differences of input concentrations between D1 and D2.

(C) BLI sensorgrams showing the RGG peptide binding to D1 or poly(U) RNA by BLI. The experiments were performed similarly to B, with the exception of D1 and RNA on the sensor tips and the RGG peptide in solution. Opaque solid lines are experimental data and dotted lines are model fits.

- Because overall, RNA binding via the RGG repeats still stimulates phase separation of the full length protein the final sentence of the abstract is a bit misleading: while factually correct (there is an RNA-independent contribution) it ignores the fact that RNA does contribute. Hence a phrase like "...enhanced by RGG repeats both dependent and independent of RNA" would be a more faithful reflection of the results.

Response: While we agree that RNA enhances phase separation *in vitro* (and include that possibility in the new Fig. 7), a major finding of our study is that PGL-3 can phase separate in the absence of RNA by relying on interactions involving structured domains and RGG repeats – we emphasize this point in the last sentence of the abstract: "These findings support an alternative model for PGL-3 PS that does not require RNA and is driven by oligomerization of structured domains that interact with RGG repeats". We have added a new section in the Discussion that expands on this on this point. Prior models assumed that PGL-3 PS depends on RNA *in vivo* because the c_{sat} of PGL-3 tagged with 6xHis-eGFP measured *in vitro* appeared higher than estimates for PGL-3 concentration in cells. Our observations use untagged PGL-3 which has a c_{sat} within the range of PGL-3 concentration *in vivo*.

Referee #3:

Kuroiwa et al. review August 2025

The authors present a well-motivated study: while the canonical drivers of phase separation (PS) are intrinsically disordered regions (IDRs) and low-complexity regions (often also disordered), emerging evidence indicates that ordered regions can also contribute-and in some cases may be the primary drivers. Using the *C. elegans* P-granule protein PGL-3 as a model, the authors employ an *in vitro* reconstitution approach to define which regions are necessary and/or sufficient for PS and how these regions interact to promote or inhibit the process. Contrary to prevailing dogma, they find that PGL-3's ordered domains (D1-D2) are not only sufficient but appear to be the principal drivers of PS in this system. The low-complexity RGG region modulates the critical concentration for PS and subtly alters condensate biophysical properties-full-length PGL-3 and D1-D2 condensates differ in viscosity, partly explained by the RGG addition experiments. The IDR linking D1-D2 exerts an inhibitory effect, as D1-D2-IDR constructs do not show PS (but the IDR does appear to multimerize in Figure 5A). Although all data are derived from *in vitro* systems (as noted in the limitations section), this work compellingly shows that-contrary to predictions from current algorithms-ordered regions can be dominant drivers of PS, suggesting that these algorithms may need revision.

The work is clearly presented and addresses an important conceptual gap. The data are of high quality, though the discussion could be strengthened by deeper mechanistic interpretation and integration of certain results that are currently underexplored.

Response: We appreciate the constructive comments that helped us refine our manuscript. We address all the suggestions point-by-point below.

Major Comments

1. Mechanism of D1-D2 Oligomerization

The manuscript states that "D1 and D2 both have the ability to form oligomers and provide the requisite multivalency for condensation," but the source of this multivalency is not elaborated. Could the data suggest that a valency greater than two is sufficient for PS, with structured domains achieving this via interdomain hydrophobic-core interactions? Incorporating Structural modeling (e.g., AlphaFold-Multimer), possibly highlighting interacting residues from the crosslinking mass spectrometry data, could strengthen the argument. My exploratory simulations suggest that D1-D2 may be capable of forming hollow shells of variable stoichiometry.

Response: We thank the reviewer for this excellent suggestion. We used AlphaFold3 to generate tetramer and octamer models for PGL-3(D1-D2) (Response figure 6A; now Appendix Fig. S4A). For each model, we measured the C α -C α distances of crosslinks observed at 125 mM NaCl in the XL-MS analyses focusing on crosslinks incompatible with published dimer structures. These analyses revealed that the tetramer model is consistent with the homodimer crosslink observed by XL-MS at K5, but not at S272 or K277 (Response figure 6B; now Appendix Fig. S4B). The octamer model was consistent with homodimer crosslinks at S272 and K277, but not at K5 (not shown). We note that the tetramer model scored medium to high in paired alignment errors (PAE) and the octamer model yielded even higher PAEs, thus these models are highly speculative (Response figure 6C, magenta squares; now Appendix Fig. S4C).

Response figure 6. AlphaFold3 prediction of D1-D2 oligomers

(A) Snapshot of an AlphaFold3 model of a PGL-3(D1-D2) tetramer. The colors highlight the four D1-D2 chains (A:green, B:cyan, C:orange, D:yellow). Magenta lines show the Ca-Ca distance between K5 in D1 of chains A:B (26.0 Å) and C:D (26.7 Å).

(B) Close-up snapshot of the D1:D1 interface from the model shown in the dotted square box in A. The canonical D1:D1 dimer model (grey) is superimposed to highlight the use of different interfaces between the two models. The magenta line shows the Ca-Ca distance of the K5 dimer crosslink for chains A and B of the AF3 tetramer model, and the yellow line shows the same for the SWISS model based on the crystal structure reported in (Aoki et al., 2021). D2 domains and chains C and D are not shown for clarity.

(C) Paired alignment errors (PAE) plot of the AF3 model shown in A. Magenta squares highlight the areas of D1 pairs that led to K5 dimer crosslink within 30Å.

I was also intrigued that the condensates in the movie appeared to exhibit a strikingly uniform size distribution, suggesting a potential self-limiting assembly mechanism.

We apologise for the confusion on droplet size in the movie. Magenta circles indicate positions of beads, not the droplets, and the entire field of view of each video is inside a

single droplet. We added an explanation to the figure legend to clarify this point. For more information on the droplet size distribution, please refer to Response figure 1, in which we observed larger and more abundance condensate of FL than D1-D2, as expected.

2. Interpretation of Figure 5A

The data in Figure 5A support an interaction between D1-D2 and the RGG region, but the behavior of the IDR in this assay is not addressed. The IDR appears to form high-order multimers that may overlap with D1 or D2 bands, yet in the LLPS assays, the IDR does not self-assemble or co-assemble with D1-D2. This discrepancy warrants discussion.

Response: We thank for raising an important point of discussion. We see that there are two main points here; the discussion around oligomerisation of the internal IDR, and the discussion around interactions between D1-D2 and the internal IDR.

To the first point, the internal IDR crosslinks to form high molecular species as the reviewer points out. We note that crosslinkers fix complexes in time and therefore distort information on timescales and strength of interactions. Due to the lack of condensation by the internal IDR by itself, we interpret that although the internal IDR of PGL-3 is capable of transient interactions, these interactions are likely too weak to support phase separation by themselves. We included this point in the revised manuscript: "XL-MS and FA crosslinking revealed few interactions between the internal IDR and D1-D2, the IDR was slightly depleted from D1-D2 condensates, and D1-D2-IDR underwent phase separation at a higher c_{sat} than D1-D2. These data suggest auto-inhibitory mechanism through the binding between D1-D2 and the internal IDR is unlikely and we interpret that the internal IDR could be acting as a passive solubilizer."

To the second point, the oligomer bands of the internal IDR indeed overlap with D1 or D2 oligomers (Fig. EV4D). However, we think this is due to their similar molecular weight and not molecular interactions. If they did interact, we would expect novel bands to appear when comparing single protein conditions and mixed conditions, and we should be able to detect them given the similar but distinct molecular weights. Experimentally, we do not detect such changes in migration patterns when the internal IDR was mixed with D1 or D2. As such, we do not expect D1-D2 and the internal IDR to co-condense, and our view is that there is no discrepancy as suggested.

3. IDR-Mediated Inhibition of PS

The mechanism by which the IDR inhibits D1-D2-driven PS is not explored. An interpretation of potential inhibitory mechanisms would strengthen the discussion.

Response: XL-MS and FA crosslinking data showed there are few interactions between D1-D2 and the internal IDR. In addition, the ability of D1-D2-IDR to phase separate, although at a higher c_{sat} , and the minimal change in D1-D2 phase separation upon addition of the internal IDR suggest that an auto-inhibitory mechanism through binding is unlikely. The internal IDR remained soluble at all conditions we tested and therefore we infer that the internal IDR is a passive solubilizer in the context of D1-D2-IDR. We expanded the discussion section to include these points.

4. Depth of Discussion

The discussion section largely reiterates results rather than interpreting them. Incorporating more mechanistic insights, speculative models, or connections to the broader PS field would enhance impact. Some experimental findings go undiscussed and should be addressed. Given that RGG motifs can function in trans, could P-granule proteins serve as RGG donors for other constituent proteins that cannot phase separate using structured domains alone? This possibility has interesting physiological implications that merit discussion.

Response: We thank the reviewer for the comment. We expanded the Discussion to include the suggestions above and also to discuss a recent paper from the Hyman group that is based on the assumption that RNA stimulates PGL-3 PS. Lewis NS, Zedlitz S, Ausserwöger H, McCall PM, Hubatsch L, Nusch M, Ruer-Gruß M, Hoegel C, Jülicher F, Eckmann CR, Knowles TPJ, Hyman AA. A mechanism for MEX-5-driven disassembly of PGL-3/RNA condensates in vitro. Proc Natl Acad Sci U S A. 2025 May 20;122(20):e2412218122. doi: 10.1073/pnas.2412218122. Epub 2025 May 12. PMID: 40354522; PMCID: PMC12107180. <https://pubmed.ncbi.nlm.nih.gov/40354522/>.

Minor Comments

1. Include scale bars in all microscopy images.

Response: We exchanged all scale bars with larger and thicker scale bars to increase visibility.

2. Clarify why crowding agents were not used in the in vitro assays.

Response: Crowding agents were not used as in previous studies^{1,2}. We added this explanation to the relevant method section.

3. In the abstract (lines 9-10), consider rephrasing "D1-D2 is oligomeric" to "D1-D2 has the potential to self-assemble."

Response: We appreciate the suggestion. We changed the phrasing to D1-D2 oligomerizes.

4. The nomenclature for the "IDR" and "RGG" regions is inconsistent and potentially misleading, given that both are disordered. It may be clearer to refer to them as regions rather than domains, reserving the term "domain" for folded, independently stable units.

Response: We thank the reviewer for this suggestion. We now use "the internal IDR" and "RGG region".

5. Correct the text at line 116 to refer to Fig. 2C; Figure 2D is not cited.

Response: We thank the reviewer for the comment. The text at line 125 (formerly line 116) correctly refers to Fig. 2B, not Fig. 2C. Fig. 2D is cited at text line 137-138.

6. Expand the legends for Figures 3B and 5D to provide sufficient methodological detail.

Response: We apologise for the insufficient explanation for these experiments. We added additional details to the figure legends.

7. For Figure 5A, clarify whether 1% crosslinking is required to observe multimerization and whether native PAGE could serve as an alternative approach.

Response: We appreciate this suggestion. Response figure 7 (now Fig. EV4B in the manuscript) suggests the oligomerisation of PGL-3 D1 and D2 using blue-native PAGE. We observed that D1 runs mostly as monomer and exhibits smearing. D2 runs mostly as monomer and dimer, and faint additional high molecular bands, but also shows smearing. The smearing migration pattern indicates unstable complex formation, consistent with fast binding kinetics.

Response figure 7. Blue-native PAGE, visualized by silver stain, of single-domain constructs, D1 and D2, of PGL-3 at indicated concentrations.

8. D1 requires D2 to PS since D1-D2 phase separates in Fig 1D while D1 and D1-IDR-RGG cannot, which suggests that D1:D2 is the minimum necessary interaction for PS with RGG interaction serving a complementary role.

Response: We thank the reviewer for this comment. We presented that D1-D2 is necessary and sufficient for PS, and we described that as likely contributing interactions, there are D1:D1 and D2:D2 and D1:D2 interactions while D1-D2:RGG interactions confers additional interactions as the reviewer mentioned. However, we did not explore specifically which interactions within D1-D2 are necessary for PS, except to test in vitro the previous finding that D1:D1 dimerization is required for PS³. We did not make mutations to block D1:D2 interactions and cannot conclude D1:D2 is the minimum necessary interaction for PS.

Dear Geraldine,

Thank you once more for the submission of your revised manuscript to EMBO Reports. As you know, it was seen again by Referee #1 and #3 who both support publication without further revision. I have now completed all checks from the editorial side and kindly ask you to address the following points, before we can proceed with the official acceptance:

- The Disclosure and competing interests statement needs to go after the Acknowledgments.
- Materials and Methods should be Methods
- References: please use et al after 10 author names; DOIs should only be used for preprints and datasets that have not been published yet.
- Jelenic et al, 2022 and Lotthammer et al, 2024 are preprints. Please add the prefix "preprint" to the in-text citations (preprint: Jelenic et al., 2022) and the label [PREPRINT] at the end of the reference in the reference list.
- Please upload the figures as individual, high resolution files (main and EV figures).
- The Appendix file is pixelated and needs to be reformatted at a higher resolution.
- The Nomenclature of EV figures should be Figure EV1, etc. instead of Expanded Figure EV1, etc.
- The following figure callouts need correction - Fig. S2C, Fig. S5E (if these are Appendix figures, then "Appendix" should be added).
- Dataset S1: the legend needs to be removed from the Appendix file and provided in the Excel sheet in the column descriptions tab. The correct nomenclature is "Dataset EV1". Please correct the file name, the name in the legend and the callout to Dataset EV1.
- Appendix Figure S5B: please provide a scale bar and define its size in the legend. Please define the box blot (max, min, whiskers etc) and specify the number and nature of the replicates (also for S5C).
- Table S1 in the Appendix file needs to be renamed to Appendix Table S1.
- All materials and methods need to be part of the main manuscript text. Please move the Appendix methods to the main methods section.
- Please remove the movie legend from the manuscript file and provide it separately as a simple README.txt file. Then the movie and the legend need to be zipped up so that we have one zip folder called Movie EV1.
- Please provide the synopsis image in jpeg, TIFF or png format and at a size of 550 pixels wide x 200-600 pixels high.
- Source Data: the SD checklist should be uploaded separately (file type: Related Manuscript file) and each Figure folder needs to be uploaded separately. Please provide the source data for Figure 6B and 6C in separate subfolders and indicate its presence in the checklist
- We perform a routine image integrity check on all revised manuscripts and noticed the following:
 - a) Figure EV1. C and E. There appear to be similar backgrounds and defining features within the figures. If you look e.g., at Figure EV1C, the background pattern is exactly the same for all four images (the dark grey spots on the light grey background). The same is true for EV1E. Could you please explain what this background represents? Is it from the imaging setup?
 - b) Could you please provide the raw imaging data for Figure 6B for a double-check of what appears to be "empty", i.e., devoid of signal in the first row? Thank you.
 - c) The images in Appendix Figure S2B for D1-D2 at 10 uM PGL-3: 200 mM and 125 mM NaCl look very similar to each other. Could you please check and clarify?
- Please provide the specific URLs for PXD064592 in the data availability statement.
- Please provide the database name for the S-BIAD2412 dataset (BioImage Archive) in the data availability statement.

- Please address the following comments in the figure legends:

a) Please provide the exact p values in the legend of figure 2E.

b) Please define the box plots in terms of minima, maxima, centre, bounds of box and whiskers, and percentile in the legends of figures 2E, 3B, 5C-E; 6C, EV5 A.

c) Please define the error bars in the legends of figure 2D.

With kind regards,

Martina

=====

Referee #1:

The authors did a great job with their revisions. I fully support publication of this manuscript without any further adjustments.

Referee #3:

The authors did a good job on the revisions for this paper. All my comments have been addressed.

* Figure EV1. C and E. There appear to be similar backgrounds and defining features within the figures. If you look e.g., at Figure EV1C, the background pattern is exactly the same for all four images (the dark grey spots on the light grey background). The same is true for EV1E. Could you please explain what this background represents? Is it from the imaging setup? - See attached screenshot.

As described in the first editorial revision, the background pattern comes from the dirt in our microscope and does not come from the samples. We also included the explanation in the Fig. EV1C legend. Please let us know anything specific that is unclear and we would be happy to address this point.

* Thank you for providing raw imaging data for Figure 6B. Could you please explain the panel in the top row that appears to be "empty", i.e., devoid of signal. - See attached screenshot.

1. The top row is at a condition without RNA as indicated in the label. The middle pane is an RNA channel. Because there is no RNA in the solution, we observe only low-level noise signals. The fluorescence intensities are levelled across conditions, and such low noise signals do not show.

2. In the attached screenshot (in the email) we see a white box in the middle pane of the top row; however, it is meant to be a black image (below). We suspect the white box is just an editing on your end, but we wanted to make sure this is the case, and we correctly raised a point above.

Geraldine Seydoux
Johns Hopkins University
Dept. of Molecular Biology and Genetics
Johns Hopkins University
School of Medicine
Baltimore, 725 N. Wolfe Street / 515 PCTB USA-Baltimore, MD 21205-2185
United States

Dear Geraldine,

I am very pleased to accept your manuscript for publication in the next available issue of EMBO reports. Thank you for your contribution to our journal.

You may qualify for financial assistance for your publication charges - either via a Springer Nature fully open access agreement or an EMBO initiative. Check your eligibility: <https://link.springer.com/journal/44319/how-to-publish-with-us>

Best regards,

Martina

>>> Please note that it is EMBO Reports policy for the transcript of the editorial process (containing referee reports and your response letter) to be published as an online supplement to each paper. If you do NOT want this, you will need to inform the Editorial Office via email immediately. More information is available here: <https://link.springer.com/partners/embo-press/editorial-policies#Peer%20review>